# WHAT ARE YOU DOING? A CLOSER LOOK AT CONTROLLABLE HUMAN VIDEO GENERATION

Our benchmark: *What Are You Doing?* (WYD)

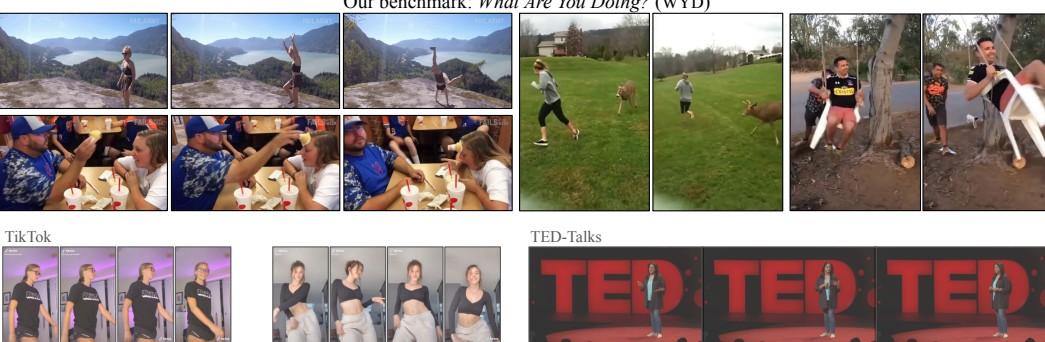

TikTok                TED-Talks

Figure 1: Samples from our WYD dataset (above) and existing datasets for controllable human video generation (below). WYD is significantly more diverse in number of actors, actions, interactions, scenes, and camera motion.

## ABSTRACT

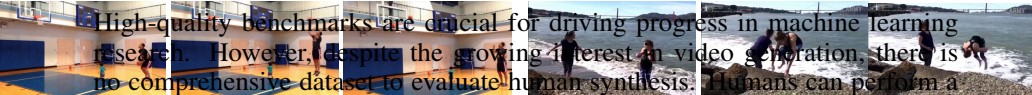

High-quality benchmarks are crucial for driving progress in machine learning research. However, despite the growing interest in video generation, there is no comprehensive dataset to evaluate human synthesis. Humans can perform a wide variety of actions and interactions, but existing datasets, like TikTok and TED-Talks, lack the diversity and complexity to fully capture the capabilities of video generation models. We close this gap by introducing 'What Are You Doing?' (WYD): a new benchmark for fine-grained evaluation of controllable image-to-video generation of humans. WYD consists of 1,544 captioned videos that have been meticulously collected and annotated with fine-grained categories. These allow us to systematically measure performance across 9 aspects of human generation, including actions, interactions and motion. We also propose and validate an evaluation framework that leverages our annotations and reflects well human preferences. Equipped with our dataset and metrics, we perform in-depth analyses of state-of-the-art open-source models in controllable image-to-video generation, showing how WYD provides novel insights about their capabilities. We release our data and code to drive forward progress in human video generation.

## 1 INTRODUCTION

Video generation has seen explosive progress in recent years, with state-of-the-art generative models demonstrating unprecedented capabilities in creating photorealistic and coherent video sequences from text prompts (Polyak et al., 2024; Wan et al., 2025; Gao et al., 2025). Nevertheless, generating humans is exceptionally hard as models must contend with the sheer complexity and high dimensionality of realistic human motion and appearance, including non-rigid body deformations and nuanced world interactions. To drive progress in this critical domain, this paper introduces WYD: a new benchmark to comprehensively and systematically evaluate model performance against the inherent challenges of human fidelity, temporal stability, and fine-grained human motion.

We specifically focus on pose-controllable generation, given the complexity of precisely describing human movements in words and its practical applications (*e.g.*, simulating a risky stunt in a movie, or making your uncle follow a dance). Generally, controllability is critical for generative models, as it enables artists to precisely specify *how* a generative model creates content, by conditioning it spatially (*e.g.*, via poses, masks, depth and edge maps; Jain et al. 2024; Ma et al. 2024; Wang et al.

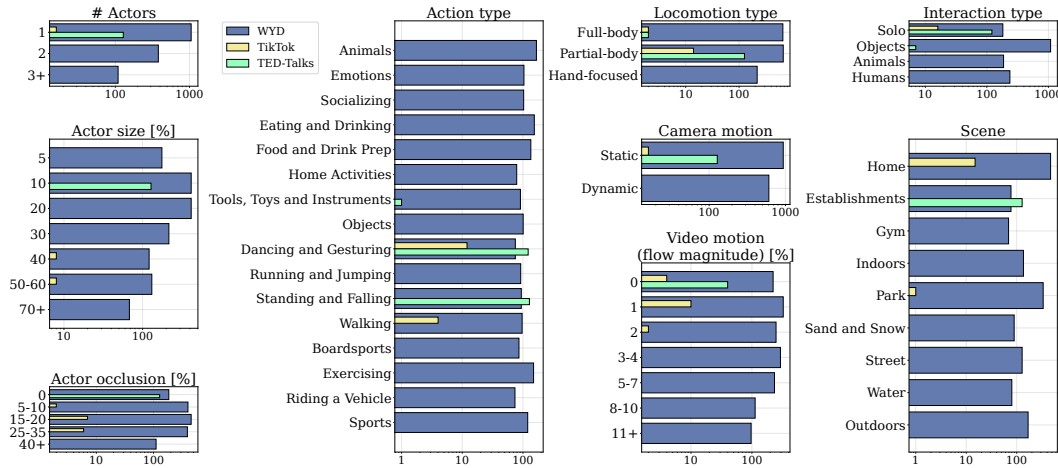

Figure 2: **Diversity of WYD**. WYD contains manually-labeled fine-grained annotations for 9 categories and 56 sub-categories relevant to human video generation. Manually annotating TikTok and TED-Talks, we verify their distinct lack of diversity across all categories. Hence, WYD serves as a more rigorous and thorough benchmark.

2024e) and temporally (*e.g.*, via motion vectors and camera positions; He et al. 2024a; Shi et al. 2024).

As the human brain is an innate expert at recognizing biological motion and appearance, achieving high-quality, controllable human generation stands as a critical feature of generative models. However, this area has received limited attention so far, and the community typically relies on two datasets: TikTok (Jafarian & Park, 2021) and TED-Talks (Siarohin et al., 2021). Both of them have limited size (Tab. 1) and a *narrow scope*, such as a single person dancing or talking in a static shot (see Figures 1 and 2). These make them insufficient to assess the full capabilities of existing models.

To bridge the gap between model capabilities and what evaluations can capture, we introduce '*What Are You Doing?*' (WYD): a new dataset of complex, dynamic videos with a wide variety of human appearance, actions and interactions. WYD includes diverse and high-quality videos filtered semi-automatically, and rigorously finalized through manual annotations (2500+ annotation hours). In particular, we built WYD considering the typical setup of controllable image-to-video generation (*e.g.*, animating a person with specific movements; Chan et al. 2019; Liu et al. 2019; Ren et al. 2020; Zhang et al. 2022; Bhunia et al. 2023; Ni et al. 2023; Karras et al. 2023; Wang et al. 2024b;d; Xu et al. 2024; Kim et al. 2024) by ensuring human actors are clearly visible in the first frame.

In addition, we argue that a detailed evaluation of model capabilities across fine-grained categories is needed to pinpoint development areas and to compare different models, but this level of detail is missing from current benchmarks (Jafarian & Park, 2021; Siarohin et al., 2021). We label the 1,544 samples in WYD according to nine categories, which enable systematic evaluations of key video-level and human-specific aspects of video generation, and make WYD significantly larger and broader than existing benchmarks (as shown in Tab. 1 and Fig. 2).

To reliably measure the ability to generate videos of humans, we propose an evaluation protocol that spans key aspects of video generation (*i.e.*, video quality, per-frame correctness, and video motion) as well as human-centric ones (*e.g.*, pose adherence). We enable human-specific metrics by customizing existing metrics on manually-curated segmentation masks for the *human actors* (see Fig. 3 in § 4).

As a representative use case of our benchmark and evaluation protocol, we investigate eight families of video generation models, including models conditioned on pose (Xu et al., 2024; Chang et al., 2024; Zhang et al., 2024b; Peng et al., 2024; Jiang et al., 2025) as well as models guided via depth and edge maps (Chen et al., 2023; Lin et al., 2024a; Wang et al., 2024c). Our results from both human and automatic evaluations show that, using WYD, we can diagnose several limitations of current SOTA models, facilitated by applying existing metrics to videos belonging to the diverse and fine-grained categories unique to our dataset. These include challenges that cannot be measured with previous datasets, such as generating cross-shot or atypical movements, interactions with objects, and more.

Our contributions are threefold: (*i*) We identify the limitations of existing benchmarks for controllable human generation, and meticulously collect WYD: a large and diverse benchmark with fine-grained annotations. (*ii*) We propose an evaluation protocol with video- and human-level metrics validated with human preferences. (*iii*) We conduct system-level evaluations of eight SOTA open-source models on WYD, and show it is harder than existing benchmarks, revealing systematic limitations and narrow training data distributions, previously undetectable.

Table 1: **Overview of data statistics.** WYD complements existing datasets for human video generation with more videos, higher diversity and fine-grained categories. See Fig. 2 for more statistics.

| | TikTok | TED-Talks | WYD |
|---|---|---|---|
| # Videos (unique clips) | 16 (14) | 128 (40) | 1,544 (1,393) |
| Video duration [s] | 8.3–23.0 | 4.3–23.1 | 1.5–15.0 |
| Video aspect ratio | portrait | landscape | portrait and landscape |
| # Actors | 1 | 1 | 1, 2, 3–8 |
| Categories | N/A | N/A | # actors, actor size % occlusion, actions, scene interactions, locomotion camera and video motion |
| Additional annotations | dense poses | N/A | video segmentation masks, 2D pose keypoints |

## 2 RELATED WORK

**Controllable video generation.** Recent algorithmic and data engineering innovations led to major progress in video generation (Ho et al., 2020; 2022a;b; Blattmann et al., 2023b). This increase in quality has encouraged a shift towards controllable generation, giving users finer control and precision than possible from text alone (Zhang et al., 2023a; Ma et al., 2025). For example, providing a reference video is a far simpler input than laboriously specifying a person's precise movements in a scene via text. Specifically, for human video generation, 2D body keypoints are the most frequently used signal, offering a sparse yet easily interpretable way to condition the output (Hu et al., 2023; Wu et al., 2023; Wang et al., 2024b; Xu et al., 2024; Lei et al., 2024). Other common visual controls include depth and edge maps (Chen et al., 2023; Zhang et al., 2023b), bounding boxes (Chen et al., 2024a; Wang et al., 2024a), and camera angles (Wang et al., 2023; He et al., 2024a).

**Benchmarks for video generation.** As the field gains momentum and video generation models achieve greater maturity, researchers are starting to establish standardized benchmarks to ensure reliable advancements. For the common text-to-video task, recent benchmarks consist of a collection of text prompts, designed to evaluate the video quality and video–text alignment in open domains. For example, VBench (Huang et al., 2024), EvalCrafter (Liu et al., 2023) and Movie Gen Bench (Polyak et al., 2024) propose comprehensive prompt sets that span multiple aspects of generic video generation. T2V-CompBench Sun et al. (2025) focuses on compositionality, while StoryBench (Bugliarello et al., 2023) evaluates video generations from a sequence of text prompts. I2V-Bench (Ren et al., 2024) and AIGCBench (Fan et al., 2023) assess consistency for the image-to-video task. Notably, most of the existing benchmarks are not suitable for evaluating controllable video generation as they lack reference videos from which to extract driving signal and against which to compare. Therefore, researchers typically rely on existing video understanding datasets (Pont-Tuset et al., 2017; Soomro et al., 2012) or collect small datasets for human evaluations (Peng et al., 2024; Jiang et al., 2025).

**Benchmarks for human video generation.** Currently, the most common benchmarks for controllable human video generation are TikTok (Jafarian & Park, 2021) and TED-Talks (Siarohin et al., 2021; Wang et al., 2024b). The TikTok dataset consists of just 16 videos scraped from TikTok, where a single person, typically at home, covers a large, centered part of the video and mostly dances in place. The TED-Talks dataset contains 128 videos from 40 talks at TED events, where a single person is on stage and gesticulates while speaking on their spot. We believe these benchmarks are extremely narrow and fail to capture the richness of human actions and interactions. Therefore, in this paper, we present a new benchmark for controllable human video generation (WYD) that addresses their shortcomings. In fact, compared to TikTok and TED-Talks, WYD is orders of magnitude larger and more diverse (Tab. 1 and Fig. 2), going beyond the status quo of static, single-person close-up videos.

**Metrics for human video generation.** Previous work typically evaluates models through pixel-level metrics, like SSIM and LPIPS, and FID (Wang, 2004; Zhang et al., 2018; Heusel et al., 2017). However, we argue that controllable human video generation requires fine-grained metrics that specifically measure alignment with the humans in reference videos. Thus, we adapt existing metrics for subject consistency (Ren et al., 2024) and motion adherence (Lin et al., 2014) to general, multi-person videos by collecting and leveraging segmentation tracks for the actors in each WYD video.

## 3 THE *What Are You Doing?* (WYD) BENCHMARK

Our goal is to benchmark the capability of video generation models to successfully synthesize human subjects in authentic, real-world settings. To effectively do so, a benchmark that satisfies different desiderata is required. Specifically, we need videos where people (*i*) are visible, (*ii*) perform a variety of actions, (*iii*) interact with each other and other objects, within (*iv*) complex and dynamic scenes.

In this section, we describe our approach to curate videos that meet these criteria from previously published datasets, and how we categorize them in order to assess different aspects typical of human video generation. We are primarily interested in human animation, given its practical applications and popularity in the field (Lei et al., 2024). We refer to our new dataset as *What Are You Doing?* (WYD).

### 3.1 DATA FILTERING

To construct a *generic* benchmark for human video generation, we require videos that capture a wide range of human activities and environments. Therefore, we rely on three publicly licensed datasets collected from Internet platforms, such as YouTube and Flickr: Kinetics (Kay et al., 2017), DiDeMo (Hendricks et al., 2017) and Oops (Epstein et al., 2020). These datasets include humans with different body poses, clothing, age and background, performing a great variety of actions (including unintentional actions that serve as an interesting testbed for atypical human motion modeling).

At a high-level, we (1) start by filtering out videos where the main actors are not humans, and (2) use a shot detector (Castellano, 2024) to remove scene cuts. We then (3) make sure that human actors are visible in the first and most of the frames using a human pose estimator (Yang et al., 2023). We further remove videos that (4) are either too short or too long (1.5–15s), whose (5) video–caption pairs have low similarity according to a fine-grained VLM (Gao et al., 2022), and (6) have low resolution. For details and examples of discarded / retained videos at each step, see App. B.

Finally, we (7) meticulously scrutinize the quality of the resulting videos and remove those with (i) significant blur, (ii) poor lighting, (iii) unstable camera, (iv) low motion, or (v) where the first frame does not capture the main actors. This process, in conjunction with the video categorization described below, was done in multiple rounds and took over 500 hours. For further quality control, the authors validated the annotations and ensured the videos' quality and high diversity, and that they would serve as a challenging and comprehensive testbed for human video generation for current and future models.

At the end of this pipeline, WYD contains 1,544 high-quality videos (from the original 18,351), which enable the fine-grained analyses, such as those described in § 5, within a tractable runtime.

### 3.2 VIDEO CATEGORIZATION

Our benchmark moves beyond the single aggregated score typical of existing datasets, aiming instead to provide a detailed, multi-faceted understanding of a model's capabilities in human video generation. Thus, we annotate our data with nine categories that capture crucial aspects of human synthesis. Each category includes sub-categories (Fig. 2), all with at least $\approx$100 samples for sufficient statistical power.

For each video in WYD, we manually label: the number of human actors; the average area they cover; the type of action they perform; the kind of body movement; how they interact with their environment; which scene they are in; and whether the camera follows them. Additionally, we use an optical flow model (Teed & Deng, 2020) to estimate the amount of video motion, and a pose detector (Yang et al., 2023) to estimate the amount of occlusion. For details, please refer to App. B.

Overall, we find that only a few categories overlap with each other significantly (see Fig. 41 in App. B). Namely, actions and interactions with animals; and video and camera motion, where high-motion videos come from dynamic camera, and videos with low motion correspond to static camera.

### 3.3 DENSE VIDEO ANNOTATIONS

In addition to labeling our videos with nine categories, we annotate each human actor in a video with tracked segmentation masks, which enable better evaluations of generated humans (§ 4). These are first obtained through SAM 2 (Ravi et al., 2025), and then manually corrected (1000 rating hours). To verify the reliability of our evaluation framework (§ 5), we also manually label (1000 annotation hours) the 2D pose keypoints for the actors in 100 representative videos. For more details, see App. B.

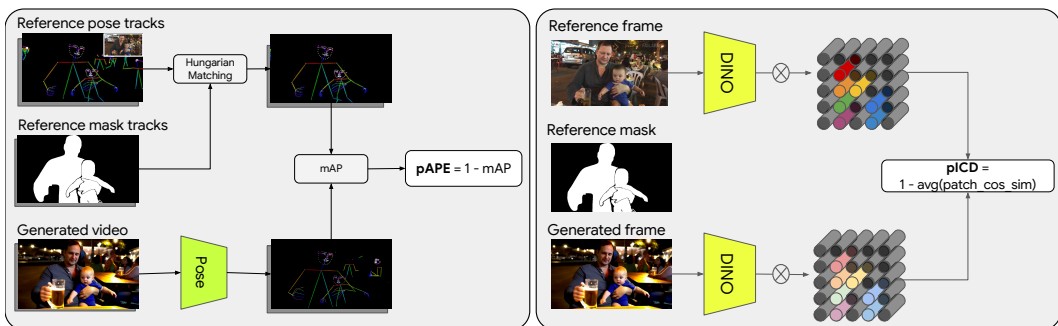

Figure 3: **Overview of our human-centric metrics.** We leverage our segmentation masks to define metrics for person consistency (pICD; left) and pose adherence (pAPE, right) that align well with human preferences (§ 6).

## 4 EVALUATION PROTOCOL

We propose the following evaluation framework (extensively validated in § 6) to evaluate model performance on WYD across four aspects: visual quality, video motion similarity, person consistency and human motion adherence. Notably, while previous work only evaluates across entire videos, we also investigate human-specific aspects by adapting existing metrics with our segmentation masks.

**Video-level evaluations.** We use the following existing metrics to measure video-level performance.

- **Video quality (FVD):** Following prior work (Ren et al., 2024; Wang et al., 2024e), we use the Fréchet Video Distance (FVD; Unterthiner et al. 2019) with an I3D backbone (Carreira & Zisserman, 2017) to compare the distribution of generated and reference videos.
- **Video motion (OFE):** Following prior work (Lin et al., 2024a; Liu et al., 2023), we compute the optical flow endpoint error (OFE) as a measure of structural dissimilarity between reference and generated videos. We use RAFT (Teed & Deng, 2020) to extract optical flows.

**Human-level evaluations.** As humans are the main focus of WYD, we adapt existing metrics to measure person-specific performance by leveraging our segmentation masks (more details in App. D).

- **Person consistency (pICD):** The ability to generate coherent human subjects is a central aspect for human video generation. Ren et al. (2024) use DINO (Caron et al., 2021) feature similarity across frames to measure subject consistency. Inspired by their results, we adapt this metric to measure human consistency. As shown in Fig. 3 (left), for each frame, we compute the average cosine similarity between the DINOv2 (Oquab et al., 2024) features of the patches corresponding to people. For consistency with other error metrics, we instead report the cosine distance and call this metric the 'person image cosine distance' (pICD).
- **Human movement (pAPE):** Another crucial requirement for controllable models is for their generated humans to strictly adhere to the underlying driving signal. We adapt the standard average precision metric (Lin et al., 2014; Ma et al., 2024) between the reference poses and those detected, using DWPose (Yang et al., 2023), in the generated video. As a scene may contain extraneous or background human subjects, we use the segmentation masks to map the poses detected in the generated video to the poses corresponding to the human *actors* (*i.e.*, the salient ones) in the reference video (Fig. 3 right). We measure AP for each video separately, and report the complement of the average AP (*i.e.*, $1 - mAP$) as an error metric for pose adherence, which we denote as the 'person AP error' (pAPE).

## 5 RESULTS

We use our new WYD dataset to investigate, for the first time, how difficult different facets of human generations are for state-of-the-art controllable video generation models (as representative models capable of precise human generation). We show that WYD allows us to pinpoint five novel failures in 8 existing models. To facilitate this, we open-source the data and the code to fully reproduce our results, which required significant resources (over 6,000 A100 GPU hours). For fair comparisons, our evaluations are at 16fps (supported by all models), and refer to this version as $WYD_{16}$.

In this section, we overlay our results by referring to specific failures of our evaluated video generation models. As visual samples require significant space, we report them in App. A for reference.

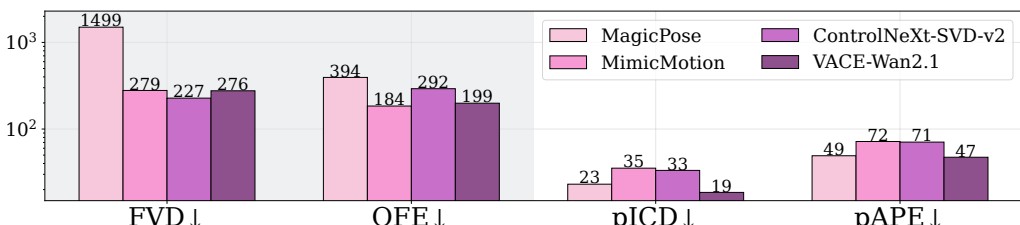

Figure 4: **Overall performance (left: video-level, right: human-level) of SOTA controllable pose-conditioned image-to-video models on WYD$_{16}$.** For VACE, we report results when conditioning on both text and poses as it results in better videos (as shown quantitatively in Fig. 10), while the other models do not support text inputs. Human generation is multifaceted and no model prevails across all metrics.

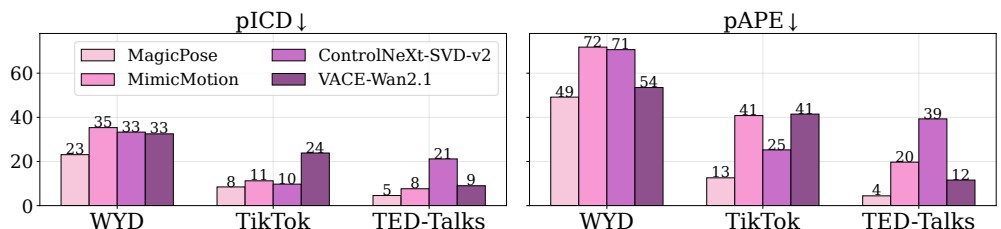

Figure 5: **Performance comparison between WYD, TikTok and TED-Talks for image+pose-conditioned models.** WYD yields larger errors, confirming that its greater diversity is more challenging for the state-of-the-art.

**Experimental setup.** Pose key-points are the most common way to condition generative models for human synthesis. They are relatively sparse, and allow artists to quickly generate humans in specific poses without having to match specific body measures (Ma et al., 2017; Zhang et al., 2023a; Wang et al., 2025). We evaluate four state-of-the-art (SOTA), open-source image-to-video models with pose conditioning: MagicPose (Chang et al., 2024), MimicMotion (Zhang et al., 2024b), ControlNeXt-SVD-v2 (Peng et al., 2024) and VACE-Wan2.1 (Jiang et al., 2025).

Besides pose-guided models, we evaluate three open-source, SOTA models that can be guided with depth or edge maps: Control-A-Video (Chen et al., 2023), Ctrl-Adapter (Lin et al., 2024a) and TF-T2V (Wang et al., 2024c). App. C additionally reports performance for text-to-video (Yang et al., 2025), image-to-video (Blattmann et al., 2023a) and dense-pose (Xu et al., 2024) baselines.

We evaluate open-source models for reproducibility, and point out that no closed-source model supports direct conditioning with poses, edge or depth maps. We also note that our evaluated models have been trained on internal datasets (or private subsets of public data) which did not include any WYD videos, as confirmed by the authors (Tab. 3 in App. A). In particular, most pose-based models have largely been trained on close-up single-person videos. While they may incur a distribution shift on WYD, we believe controllable human generation should go beyond those simple cases, and therefore propose WYD to drive progress towards more general and challenging videos.

**Overall performance on WYD$_{16}$.** Fig. 4 shows the overall performance of pose-guided models on WYD$_{16}$. We see that VACE achieves overall better performance, being on-par with MagicPose in generating *humans* (pICD, pAPE) whilst also obtaining competitive video-level performance (FVD, OFE) with MimicMotion and ControlNeXt (MagicPose obtains high FVD because of flickering).

To show the versatility of our benchmark, we analyze depth- and edge-guided models in App. C. While TF-T2V tends to outperform all other models, Control-A-Video and Ctrl-Adapter have low visual quality (FVD, pICD). Inspecting Control-A-Video's generations, for example, we find that its videos follow the underlying driving signal well but at the expenses of distorted colors and artifacts.

**WYD is harder than previous benchmarks.** In Fig. 5, we compare the errors in human metrics between WYD and TikTok or TED-Talks. WYD is consistently harder in both person consistency and movement, with error rates 1.8–4.6× higher for pICD, and 1.8–12.3× higher for pAPE. That is, WYD poses significant challenges to SOTA systems, suggesting it can be used to drive progress in this area. Moreover, in Fig. 44 (App. C), we show that the bottlenecks in WYD stem from the inherent difficulty of generation, as the models' auto-encoders can significantly better reconstruct the videos.

**Combining multiple modalities.** We study the extent to which different modalities contribute to VACE (our only multi-task model) performance on WYD. As shown in Fig. 10, adding both pose and text guidance improves model performance. As expected, pose is especially useful for better motion adherence, while detailed LLM-generated captions improve person consistency. For dense-conditioned

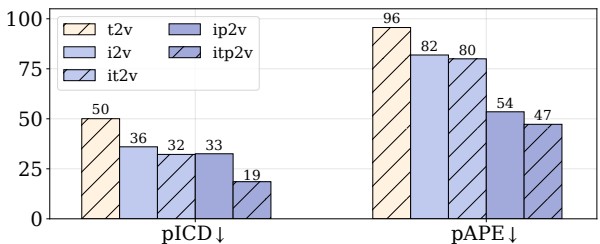

Figure 6: Performance comparison between different conditioning conditions for VACE on $\text{WYD}_{16}$. Combining multiple signals (i: image, p: pose, t: text) improves the model's performance.

models, we see that Ctrl-Adapter's generations also improve when adding text guidance (see App. C).

**Ablating pose signals.** We performed system-level evaluations above, using the pre- and post-processing used by the original authors. However, MagicPose uses the OpenPose (Cao et al., 2019) human pose estimator, whilst MimicMotion and ControlNeXt all use the more performant DWPose Yang et al. (2023), to generate pose conditioning signals from the reference video.

To investigate the sensitivity of our results to pose estimator inaccuracies, we manually annotated 100 videos (184 actors) with 2D body keypoints, and our results in App. C show that the models' performance is largely unaffected, and relative performance unchanged, thereby validating the correctness and robustness of our findings. Moreover, although replacing OpenPose with the more accuracy DWPose detector improves MagicPose's performance, it still lags behind the remaining models (App. C). Similarly, the relative ordering of models is unchanged when they all use OpenPose instead. Therefore, our findings are consistent irrespective of the pose-detector.

## 5.1 DIAGNOSING MODEL PERFORMANCE

The categories that we have collected for WYD are not only useful to ensure its richness of human videos, but can also be used to investigate model performance across these axes. In other words, we can apply our evaluation framework to different *slices* of WYD and gain insights into the strengths and weaknesses of models. This information can be used to guide model development or, as we do next, to show specific limitations of the best performing models (MimicMotion, ControlNeXt and VACE).

**Towards multi-person videos.** With WYD enables evaluating model performance for controlling multiple humans for the first time. As shown in Fig. 7, both visual quality (FVD) and pose adherence (pAPE) degrade for *2* and *3+* actors. Investigating the generations qualitatively, we find that models blend humans' appearances, especially when *humans interact* with each other (Fig. 8).

**Towards complex interactions.** FVD shows that MimicMotion struggles to generate high-quality videos of actions with *animals*. Inspecting the model's generations for this category, we see that the model tends to discard animals entirely, resulting in surreal generations (Fig. 9). This is likely because the model was mostly trained on close-up single-person videos (as confirmed by the authors).

**Towards diverse environments.** Our results (Fig. 7) show that all the models perform much better for videos taking place at *home*. Common, indoor environments (*e.g.*, *gym*) as well as more exotic, outdoor ones (*e.g.*, *sand and snow*) show large room for improvement for our benchmarked models.

**Towards dynamic videos.** Our metrics confirm that generating videos with high-level of motions are harder to generate for SOTA models. In fact, performance worsen w.r.t. both FVD and pAPE when (i) the camera is *not static*, (ii) the optical flow magnitude is larger (*e.g.*, *10%*), (iii) the actor moves their entire body (*full-body* locomotion), or (iv) they perform rapid actions (*e.g.*, *boardsports*, *running* or *riding a vehicle*). This is exemplified in Fig. 10 on a dynamic, skateboarding video.

**Towards humans of varying sizes.** While current models are mostly trained on close-up videos, we can measure model performance across humans covering different scales in WYD. Human actors that cover larger portions of the video are easier to control (lower pAPE) but the resulting videos typically have lower quality (higher FVD; Fig. 7), suggesting greater difficulty for human details.

All in all, we think that the findings that we obtain from our dataset and evaluation protocol are aligned with the experience that researchers and practitioners in the field have from using these models. We believe that being able to verify them in WYD is crucial, as it acts as an encouraging signal that we can track and improve performance across these axes through our new benchmark.

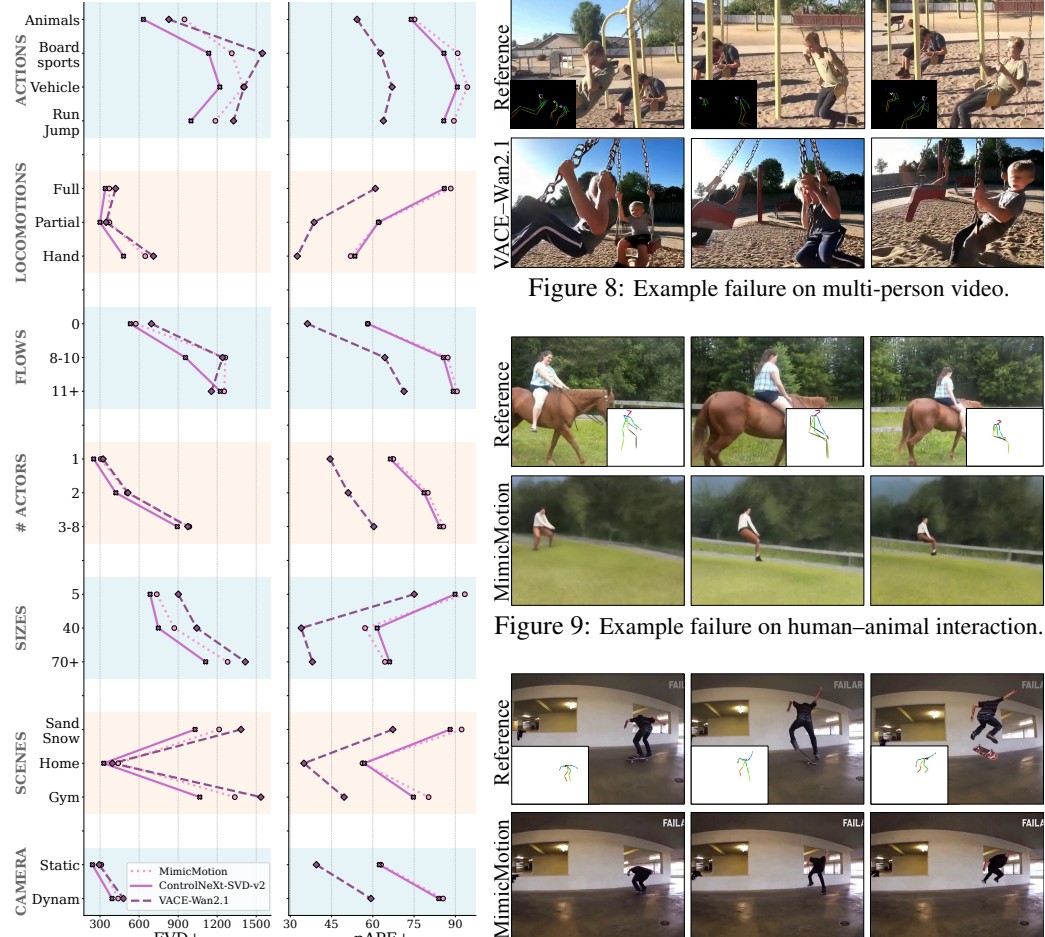

Figure 7: Performance across multiple categories.

Figure 8: Example failure on multi-person video.

Figure 9: Example failure on human–animal interaction.

Figure 10: Example failure on dynamic video.

## 6 VALIDATING AUTOMATIC METRICS

Automatic evaluation metrics are instrumental in benchmarking a model's performance. However, automatically evaluating different aspects of video generation is difficult: While multiple metrics have been proposed (Wu et al., 2024; He et al., 2024b; Wiles et al., 2025), many have been shown to have little correlation with human judgment (Bugliarello et al., 2023; Kirstain et al., 2023; Lin et al., 2024b).

Therefore, a careful validation of our proposed evaluation framework (§ 4), specifically in our context of controllable human video generation, is crucial to establishing its credibility. We do so by comparing our selected metrics with alternatives for each key generation aspect, and showing that they reflect well human preferences. Tab. 2 summarizes our results (for more details, see App. D).

**Validation setup.** We assess the performance of different metrics by aligning them with human judgments in two settings. First, we perform side-by-side model comparisons. We use four templates to assess video quality, video motion, human quality, and people movements. Second, we manually verify the model rankings given by the metrics against generated samples (see App. D for details).

**Measuring human movement via poses.** Pose AP is the gold standard metric to evaluate keypoint detection (Lin et al., 2014), and has previously been used for single-person generation (Ma et al., 2024). Our pose AP error (pAPE) adapts pose AP to robustly evaluate multi-person video generation by leveraging our collected segmentation and pose tracks (§ 4). Tab. 2 shows that pAPE agrees well (72%) on human evaluations of people's movements. Moreover, pAPE allowed us to discover that MimicMotion and ControlNeXt always re-scale and re-center the generated humans ( Fig. 11). This is part of the models' pre-processing code (Tencent, 2024; Deep Vision Lab, 2024), and may stem from the overdependence on simpler datasets with only a single, large, centered actor. This further emphasizes the need for more diverse benchmarks for controllable human generation, such as WYD.

Table 2: **Side-by-side evaluations.** We report Spearman rank correlation for video quality, and pair-wise accuracy for the rest. Our selected metrics agree with human preferences from side-by-side (SxS) studies.

| | Metric | Performance [%] |
|---|---|---|
| **Video quality** | FVD | **96.36** |
| | FID | 22.24 |
| | JEDi | **96.36** |
| | VMAF | 29.65 |
| **Video motion** | OFE | **82.10** |
| | DPT | 67.37 |
| **Human consistency** | pICD | **72.67** |
| | ICD | 67.33 |
| | RMSE | 38.55 |
| | SSIM | 62.65 |
| **Human movement** | pAPE | **71.95** |
| | pOFE | 61.45 |

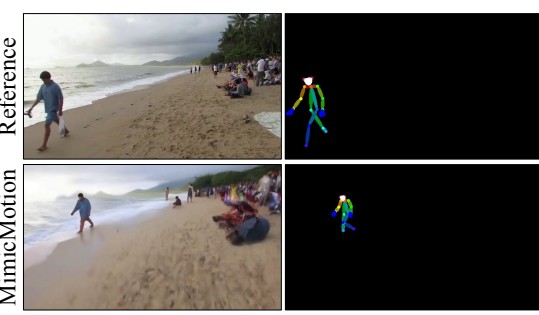

Figure 11: **Example of pose re-scaling in MimicMotion.** The pose detected in the generated video (bottom right) is re-scaled and re-centered compared to the pose from the reference video (top right). Humans are not typically sensitive to such perceptual changes but our pAPE metric is.

**pICD captures person consistency.** We evaluate the ability of visually-controlled models to produce humans that closely resemble their ground-truth references. Ren et al. (2024) use DINO (Caron et al., 2021) feature similarity across frames to measure subject consistency. Inspired by their results, we adapt this metric (ICD) to measure human consistency by leveraging our collected segmentation masks (pICD; § 4). As shown in Tab. 2, pICD is 5% more accurate than frame-wide ICD, and better captures person consistency according to human evaluations.

**FVD better measures video quality.** We compare different metrics as proxies for overall video quality: FID (Heusel et al., 2017), FVD, JEDi (Ya et al., 2024) and VMAF (Li et al., 2020). We measure Spearman rank correlation between the ranked human judgments and the scores of each metric. We find that the widely used FID and SSIM metrics (*e.g.*, Hu et al. 2023; Chang et al. 2024) are inadequate, while FVD and JEDi much better agree with humans. We opt for FVD being faster.

**Limitations of our metrics.** While our metrics show better accuracy with human preferences, we point out a few of their current limitations. Firstly, the pose estimator used in *pAPE* detects humans in each frame independently, resulting sometimes in incoherent sequences, as well as incorrect or hallucinated poses from poorly generated people. Secondly, our *video quality metric* suggests that it is easy to generate videos of human–object interactions. However, looking at MimicMotion and ControlNeXt's generations, we find that objects often disappear. As they typically cover a small percentage of the frame, metrics like FVD are largely unaffected by this, and do not reveal this issue.

In this work, we do not explicitly investigate metrics for face quality, but notice that faces are often of poor quality. This makes it challenging to evaluate *emotions*, especially when humans are small.

Finally, our evaluation protocol is based on visually-conditioned image-to-video models. This allows us to propose metrics like pICD and pAPE that better capture human-level properties via segmentation masks. We encourage further work to assess text-only generations (Sun et al., 2025).

## 7 CONCLUSION

Video generation has a tremendous potential to impact our society, and it is thus imperative to comprehensively assess model capabilities. As a milestone to this end, we collected WYD: a dataset to evaluate the synthesis of humans in real-world settings. Our analysis showed that WYD is more diverse and challenging than prior benchmarks for controllable human video generation. It is equipped with fine-grained categories that allowed us to discover several failure modes of state-of-the-art open-source technologies, through both automatic and human evaluations. By releasing WYD, we aim to move beyond the current narrow scope of close-up single-person human video generation, and to drive forward progress towards the more ambitious goal of *generic* human generation.

We focused on visually-conditioned image-to-video generation technologies in this work, as methods capable of precisely controlling human actors. A promising direction for future work in evaluating human video generation technologies would be to develop more flexible vision tools to reliably assess model performance across various aspects, following and extending the recent work in text-to-image (Huang et al., 2023) and text-to-video (Sun et al., 2025) generation.

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

App. B (pp. 34–42) provides further details from our data preparation pipeline. Fig. 27 shows a high-level overview of the filtering process. 4 and 5 list additional details of the dataset construction (p. 34), while Figures 36 and 37 compare video duration and resolution across the WYD, TikTok and TED-Talks datasets (p. 38). 28 to 34 (pp. 35–37) show sample videos that were removed as part of the filtering process, and Fig. 35 (p. 37) shows more examples from the final WYD dataset. Figures 38 to 40 (pp. 40–41) show our UIs for video categorization, segmentation and pose annotations. Fig. 41 displays the overlap in videos between any two categories.

App. C reports additional results from our experiments. Fig. 42 (p. 43) shows the overall performance of depth- and edge-conditioned models on $\text{WYD}_{16}$. Fig. 43 shows the difference in errors of depth- and edge-conditioned models when adding captions as an additional source of guidance. Figures 44 to 46 (pp. 43–44) instead provide quantitative support for the ablations of pose-guided models discussed in § 5. Moreover, Figures 47 to 50 (pp. 44–46) report and discuss category-level performance of our top-performing models (MimicMotion, ControlNeXt and VACE).

App. D includes further details about our human evaluation protocols. We report our instructions and setup for side-by-side human evaluations in p. 47, and show our UI in Fig. 51 (p. 46). Figures 52 to 54 (pp. 47–48) present and discuss how metrics that we considered to measure different aspects of video generation score our evaluated models.

Finally, we share some ethical considerations related to controllable human video generation in App. E (p. 48), where we additionally remark that our WYD dataset is meant to be used for academic research purposes only.

| Model | Condition | Extractor | Training data | Close-up single-person videos? | WYD overlap? |
|---|---|---|---|---|---|
| MagicAnimate (Xu et al., 2024) | Dense pose | Detectron2 | TikTok (Jafarian & Park, 2021) | Yes | No |
| MagicPose (Chang et al., 2024) | 2D pose | OpenPose | TikTok (Jafarian & Park, 2021) | Yes | No |
| MimicMotion (Zhang et al., 2024b) | 2D pose | DWPose | Internal | Yes | No |
| ControlNeXt-SVD-v2 (Peng et al., 2024) | 2D pose | DWPose | Internal | No | N/A |
| VACE-Wan2.1 (Jiang et al., 2025) | 2D pose | DWPose | N/A | N/A | N/A |
| Control-A-Video (Chen et al., 2023) | Depth / Canny | MiDaS / OpenCV | WebVid (Bain et al., 2021) subset* + internal | No | No |
| TF–T2V (Wang et al., 2024c) | Depth | MiDaS | WebVid (Bain et al., 2021) subset* + internal | No | No |
| Ctrl-Adapter (Lin et al., 2024a) | Depth / Canny | MiDaS / OpenCV | Panda-70M (Chen et al., 2024b) | No | No |

Table 3: **Overview of evaluated models.** We list models' conditions and used extractors, their training data and whether it mostly consists of close-up single-person videos, and whether any of the video datasets used in WYD were used by them during training. We thank the authors for clarifying information about their training data and confirming the absence of overlap with our evaluation videos. * Note that different models rely on different subsets of WebVid.

# A SAMPLES OF GENERATED VIDEOS

An overview of the evaluated models is shown in Tab. 3. Figures 12 to 26 show and discuss the limitations of samples generated by all the models for five WYD examples.

A person wearing a blue jacket goes down a snow hill on skis.

Figure 12: Example generations of our evaluated pose-conditioned models (MagicAnimate uses dense poses). We can see how people's appearance changes in MagicPose, although matching the human movements the best. We can also see the size mismatches in ControlNeXt and MimicMotion.

A person wearing a blue jacket goes down a snow hill on skis.

Figure 13: Example generations of our evaluated depth-conditioned models. We can see how people's appearance changes in TF-T2V, increasing saturation in Ctrl-Adapter and distortions in Control-A-Video.

A person wearing a blue jacket goes down a snow hill on skis.

Figure 14: Example generations of our evaluated edge-conditioned models. We can see increasing saturation in Ctrl-Adapter and distortions in Control-A-Video.

A woman wearing a white top is riding a brown horse while horses are standing on the brown surface.

Figure 15: Example generations of our evaluated pose-conditioned models (MagicAnimate uses dense poses). We note the challenges in camera motion for all models, the distortions of characters in MagicAnimate, and flickering effects in MagicPose, as well as horse disappearance in MimicMotion.

A woman wearing a white top is riding a brown horse while horses are standing on the brown surface.

Figure 16: Example generations of our evaluated depth-conditioned models. We can see increasing saturation in Ctrl-Adapter and distortions in Control-A-Video, while TF-T2V best matches the overall scene.

A woman wearing a white top is riding a brown horse while horses are standing on the brown surface.

Figure 17: Example generations of our evaluated edge-conditioned models. We can see increasing saturation in Ctrl-Adapter and distortions in Control-A-Video.

A man wearing white clothes is sitting on the floor and pouring egg mixture into a pan and spreading it.

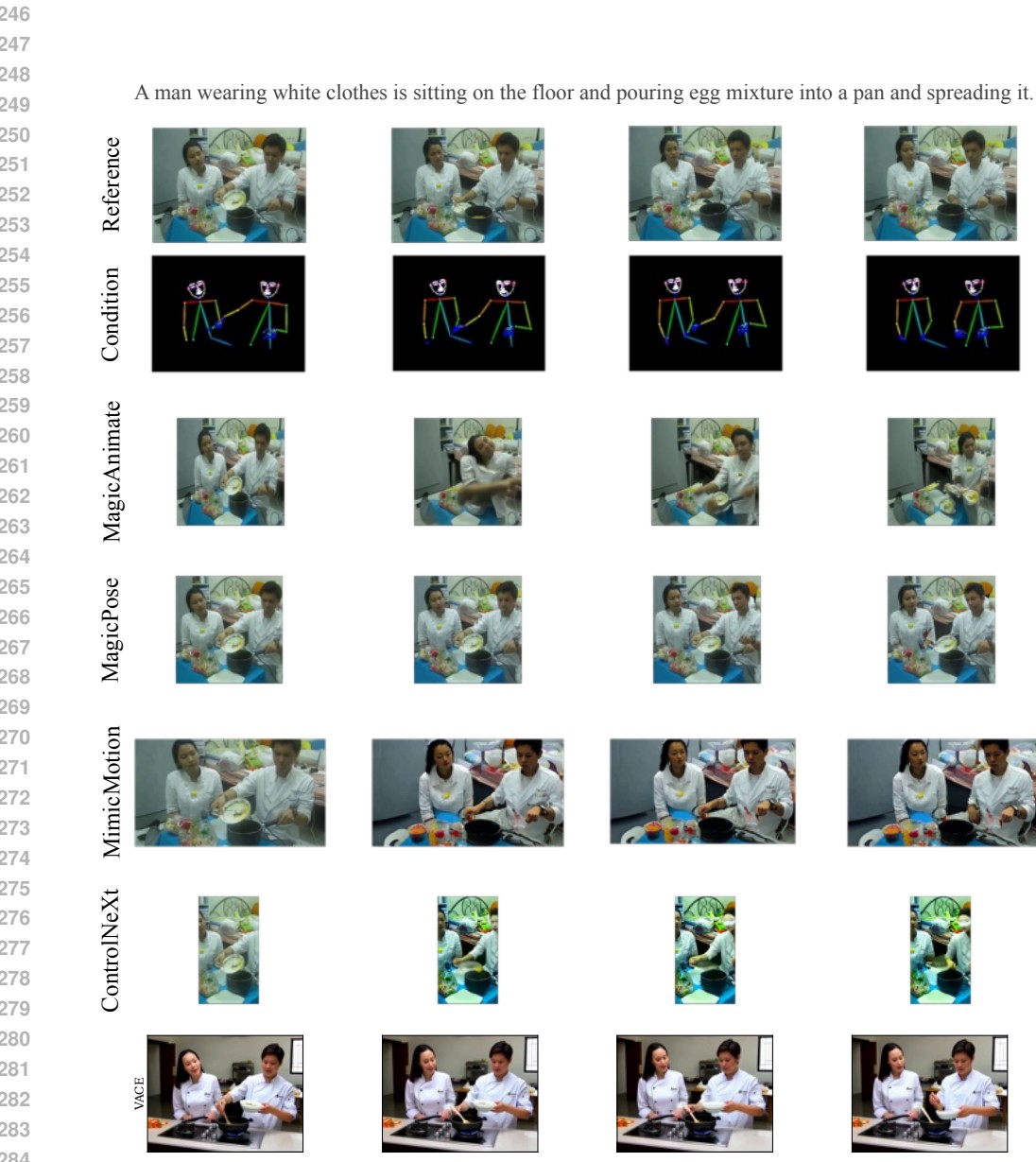

Figure 18: Example generations of our evaluated pose-conditioned models (MagicAnimate uses dense poses). We can see how MimicMotion changes the facial traits of humans towards specific age and beauty standards, and how it also fails to make the man interact with the pan. Due to its pre-processing, ControlNeXt misses the face of the man in the first frame and later creates a different one.

A man wearing white clothes is sitting on the floor and pouring egg mixture into a pan and spreading it.

Figure 19: Example generations of our evaluated depth-conditioned models. We can see how Ctrl-Adapter change the facial traits of humans towards specific age and beauty standards. We can still see increasing saturation in Ctrl-Adapter and distortions in Control-A-Video, although less than in previous, dynamic examples.

A man wearing white clothes is sitting on the floor and pouring egg mixture into a pan and spreading it.

Figure 20: Example generations of our evaluated edge-conditioned models. Similar to their depth-conditioned counterparts, Ctrl-Adapter shows increasing saturation and Control-A-Video presents distortions.

A man wearing a white t-shirt is sitting while holding a fish in his hand.

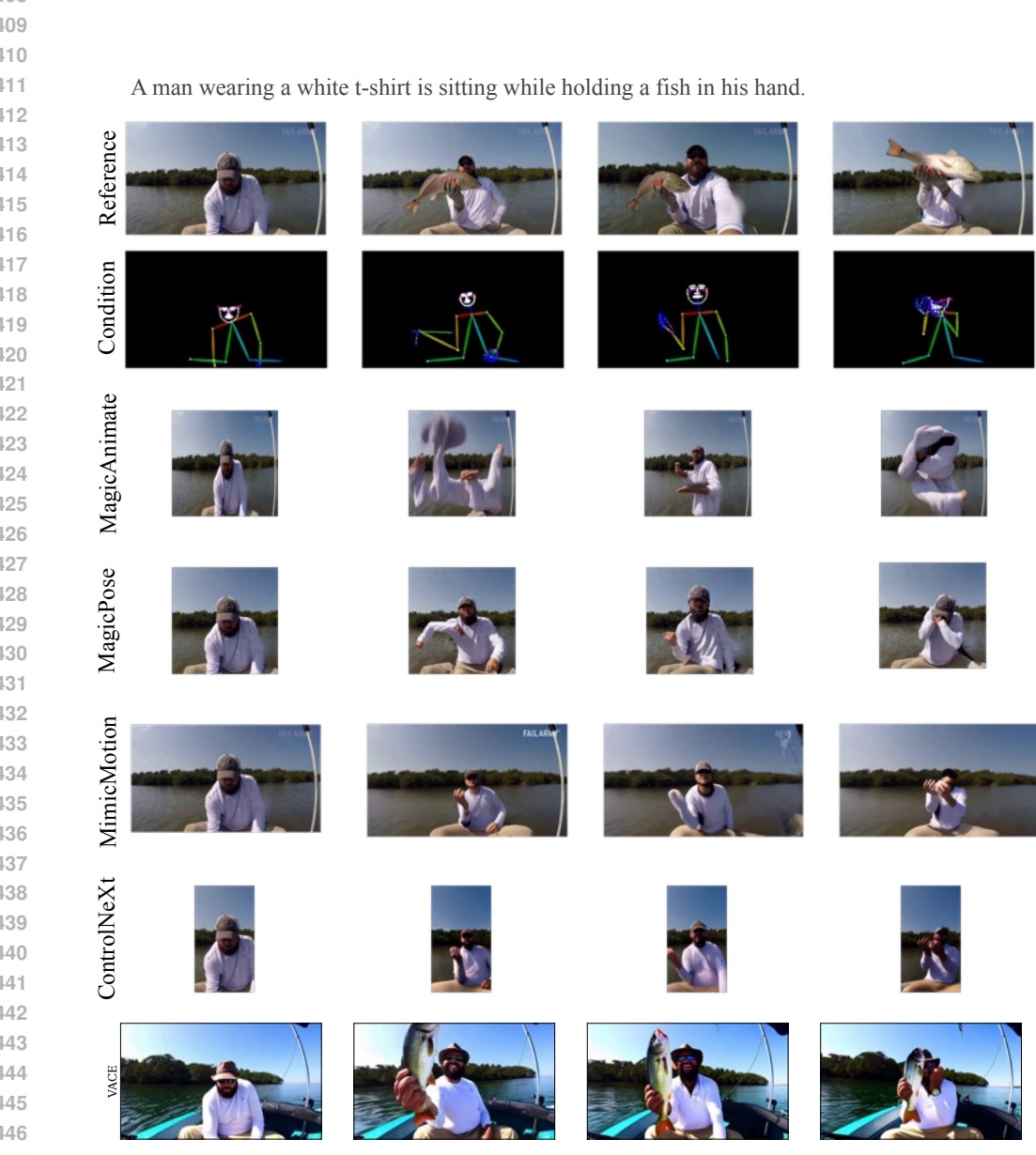

Figure 21: Example generations of our evaluated pose-conditioned models (MagicAnimate uses dense poses). Besides the large distortions and artifacts in MagicAnimate, all pose-conditioned models fail to generate the fish. MimicMotion again changes the man's appearance by removing his beard.

A man wearing a white t-shirt is sitting while holding a fish in his hand.

Figure 22: Example generations of our evaluated depth-conditioned models. We see high levels of saturation and distortions in Ctrl-Adapter and Control-A-Video, respectively. TF-T2V is the only model that synthesizes the fish well.

A man wearing a white t-shirt is sitting while holding a fish in his hand.

Figure 23: Example generations of our evaluated edge-conditioned models. Ctrl-Adapter and Control-A-Video generates videos with very high levels of saturation and distortion.

The girl hugs the man.

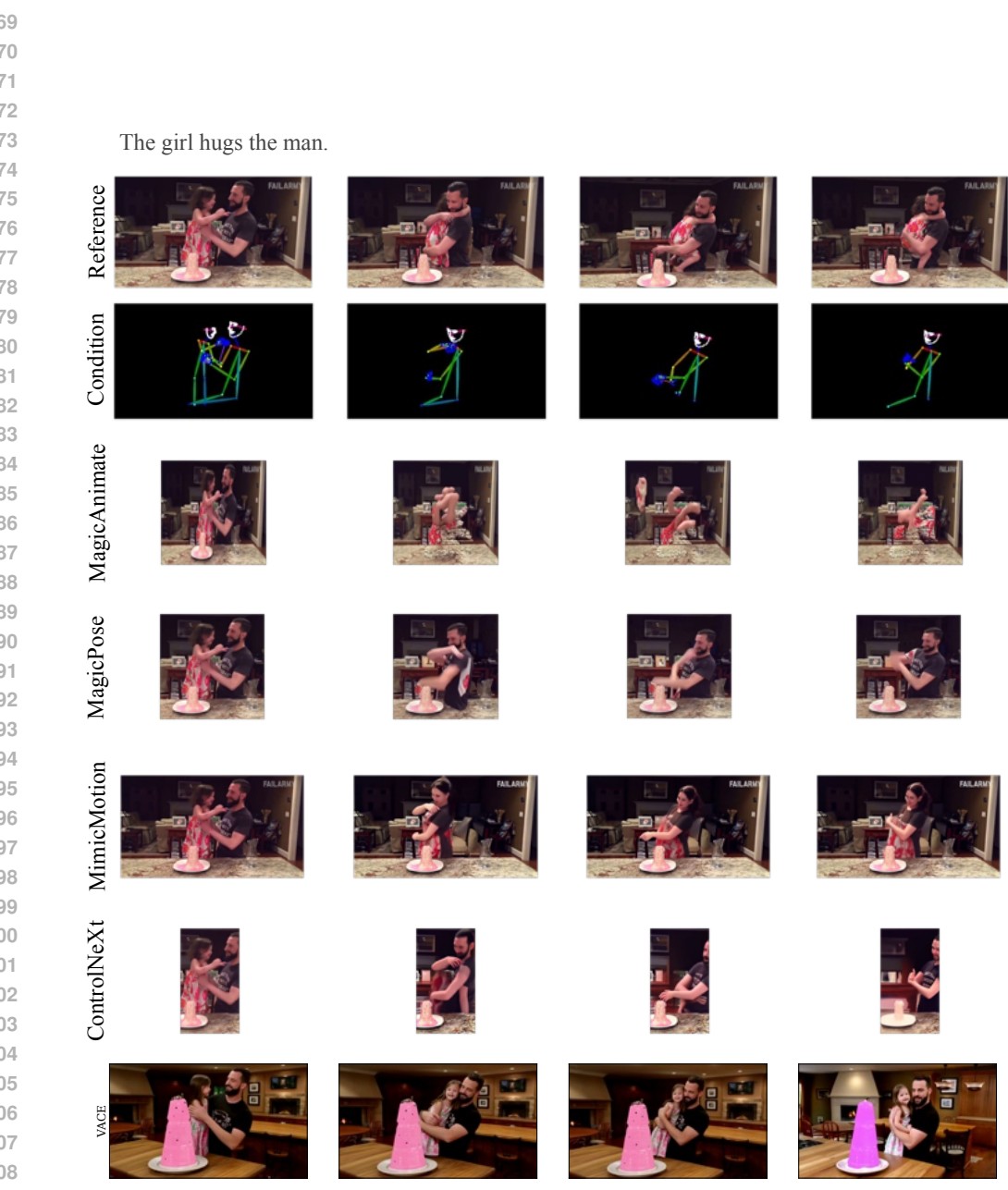

Figure 24: Example generations of our evaluated pose-conditioned models (MagicAnimate uses dense poses). All models struggle with generating multiple humans interacting with each other consistently due to the limitations of the pose extractor. For example, MagicPose and ControlNeXt make the girl disappear, while MimicMotion merges the girl and the man into a woman.

The girl hugs the man.

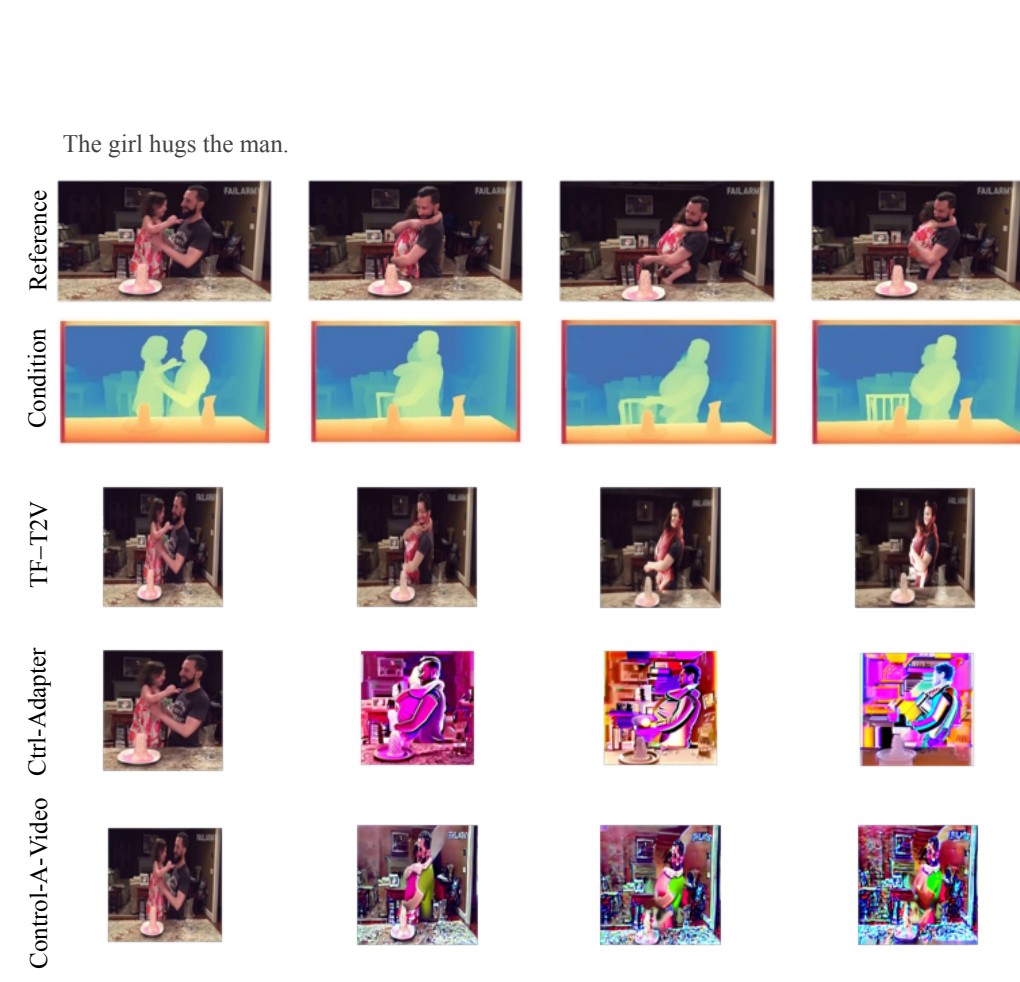

Figure 25: Example generations of our evaluated depth-conditioned models. We can see that even TF-T2V struggles to make the two humans interact with each other, resulting in a video where the man and the girl are merged into a single person with low-quality facial traits.

The girl hugs the man.

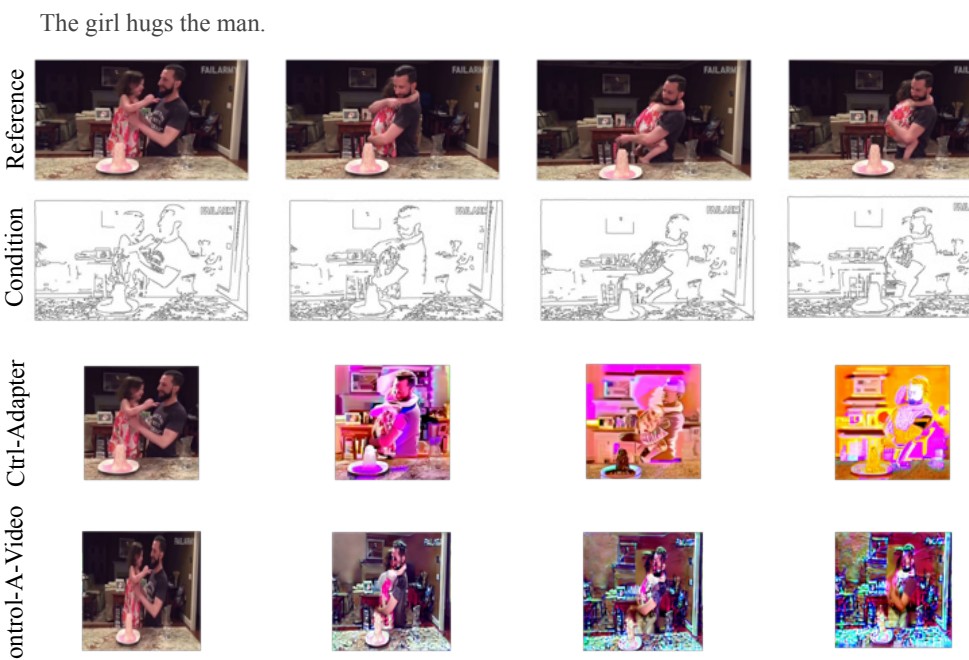

Figure 26: Example generations of our evaluated edge-based models. Once again, the generations of Ctrl-Adapter and Control-A-Video are affected by high levels of saturation and distortion, despite being a static scene.

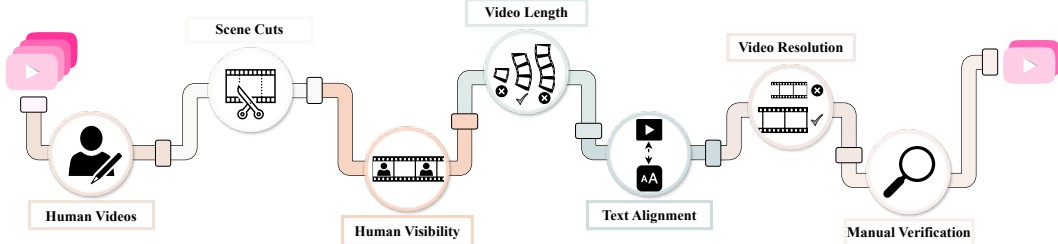

Figure 27: **WYD data filtering pipeline.** Our pipeline includes 7 steps: identifying videos with human actors, removing scene cuts, ensuring human visibility, removing short/long videos, keeping videos with high text alignment, removing low-res videos and manual verification.

| Processing step | *Dev* UVO | Oops | DiDeMo | *Test* UVO | Oops | DiDeMo |
|---|---|---|---|---|---|---|
| StoryBench | 2332 | 3472 | 2607 | 3895 | 3726 | 2319 |
| Videos of humans | 2085 | 3009 | 1479 | 3497 | 3168 | 1356 |
| Scene cuts | 1960 | 2962 | 1497 | 3264 | 3126 | 1342 |
| Human visibility | 996 | 1800 | 847 | 1638 | 1938 | 783 |
| Video length | 941 | 1545 | 706 | 1550 | 1650 | 670 |
| Text alignment | 545 | 859 | 550 | 870 | 953 | 511 |
| Video resolution | 545 | 859 | 501 | 870 | 953 | 462 |
| Manual verification (WYD) | 235 | 323 | 97 | 346 | 424 | 119 |

Table 4: Number of StoryBench entries in our datasets after each step in the WYD data preparation pipeline (see Fig. 27 and § 3.1).

| | | | | | | | | |
|---|---|---|---|---|---|---|---|---|
| Baby girl one | Bikers | Daddy | Group of People Two | Lady | Mother | Persons | Tourists |
| A baby | Boy | Dancers | Group of children | Lady one | Musicians | Pianist | Twins |
| A boy | Boy | Divers | Group of men | Lady three | Officer | Player | User |
| A girl | Boy Four | Drummer | Group of musicians | Lady two | Old lady one | Players | Woman |
| A man | Boy One | Everyone | Group of people | Man | Old lady two | Police officer | Woman |
| A old man | Boy Three | Family | Group of people | Man | Old man | Priest | Woman |
| A person | Boy Two | Fighter | Group of people one | Man Five | Old woman | Rangers | Woman One |
| A person | Boy five | Fighters | Group of people one | Man Four | Others | Referee two | Woman Three |
| A woman | Boy four | Fourth man | Group of people two | Man One | Passengers | Rest | Woman Two |
| Baby | Boy one | Gentleman | Group two | Man One | People | Rider | Woman five |
| Baby | Boy one | Girl | Guitarist | Man Three | Peoples | Rider one | Woman four |
| Baby Boy | Boy three | Girl | Guy | Man Two | Performer | Rider one | Woman one |
| Baby boy | Boy two | Girl three | He | Man five | Person | Rider two | Woman one |
| Baby girl | Boy two | Girl One | Judge | Man four | Person | Riders | Woman three |
| Baby girl one | Bride | Girl Two | Kid | Man one | Person Four | Runner | Woman two |
| Baby girl two | Bridegroom | Girl five | Kid | Man one | Person One | Second man | Woman two |
| Baby girl two | Child | Girl four | Kid Four | Man one | Person Three | Singer | Women |
| Baby one | Child | Girl one | Kid Three | Man six | Person Two | Singers | Women |
| Baby two | Child one | Girl one | Kid Two | Man three | Person five | Someone | Women one |
| Band | Child one | Girl third | Kid four | Man three | Person four | Speaker | Young man |
| Batter | Child two | Girl three | Kid one | Man two | Person one | Swimmers | kid |
| Bichon frise | Child two | Girl two | Kid one | Man two | Person one | They | kid one |
| Bicyclist | Couple | Girl two | Kid three | Marchers | Person three | Third | kid three |
| Bicyclists | Cyclist | Girls | Kid two | Mascot | Person three | Third boy | kid two |
| Bike rider | Cyclist Two | Group | Kid two | Members | Person two | Third girl | person |
| Biker | Cyclists | Group of People One | Kids | Men | Person two | Third man | woman |

Table 5: List of unique human actors extracted from StoryBench annotations.

## B    DATA ANNOTATION DETAILS

In this section, we provide additional details, statistics and samples of our data preparation and categorization processes.

**Data licenses.** We rely on three publicly available datasets to source videos for our benchmark. Kinetics Kay et al. (2017) is released under a CC BY 4.0 license, DiDeMo Hendricks et al. (2017) is released under a CC BY-NC-SA 2.0 license, and Oops Epstein et al. (2020) is released under CC

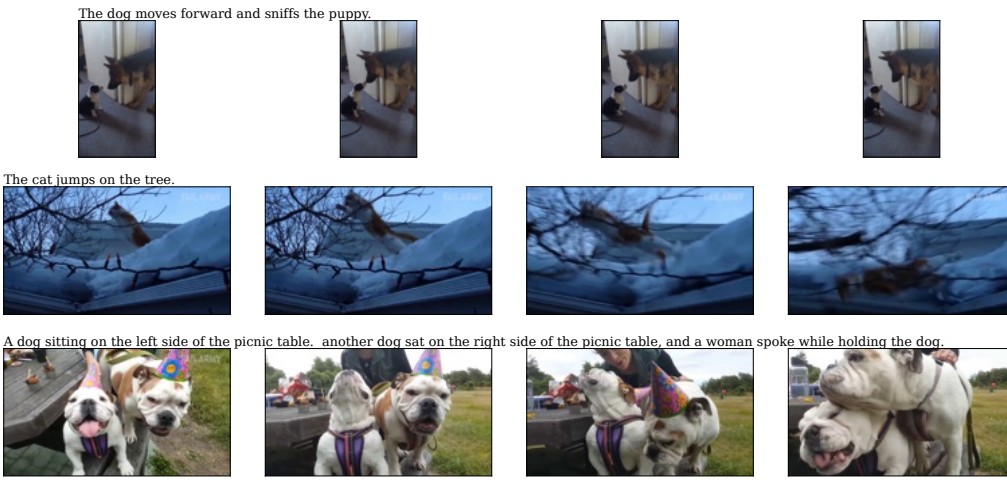

Figure 28: Discarded samples due to no human actor.

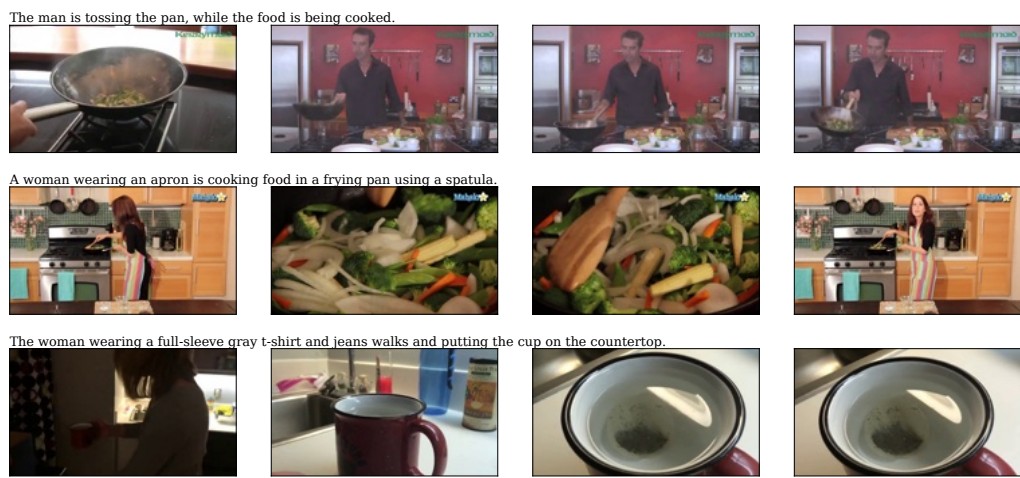

Figure 29: Discarded samples due to scene cuts.

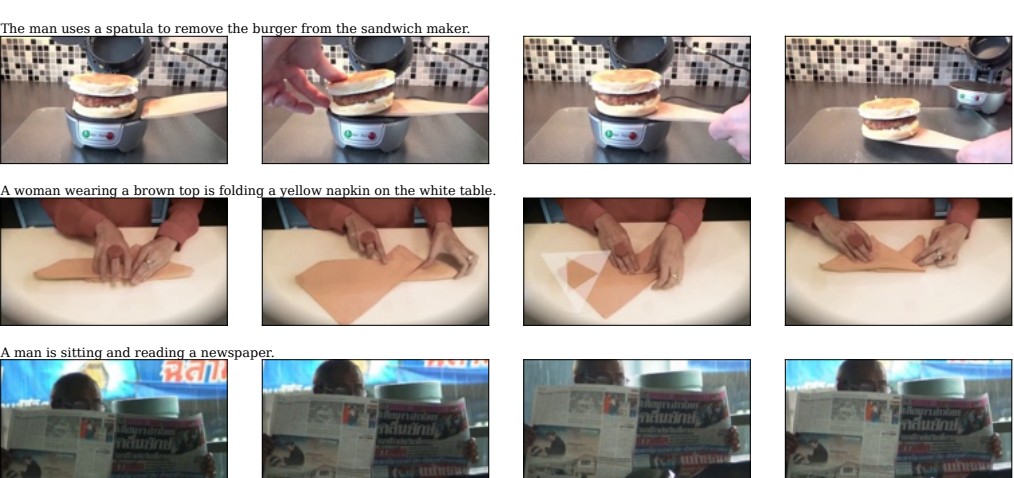

Figure 30: Discarded samples due to low human actor's visibility.

The man wearing a grey shirt and pants takes a turn, his scooter tire hits the corner of concrete road.

The man falls down.

A girl wearing a grey jacket is sitting at the back side inside the car and eating something while talking with the man.

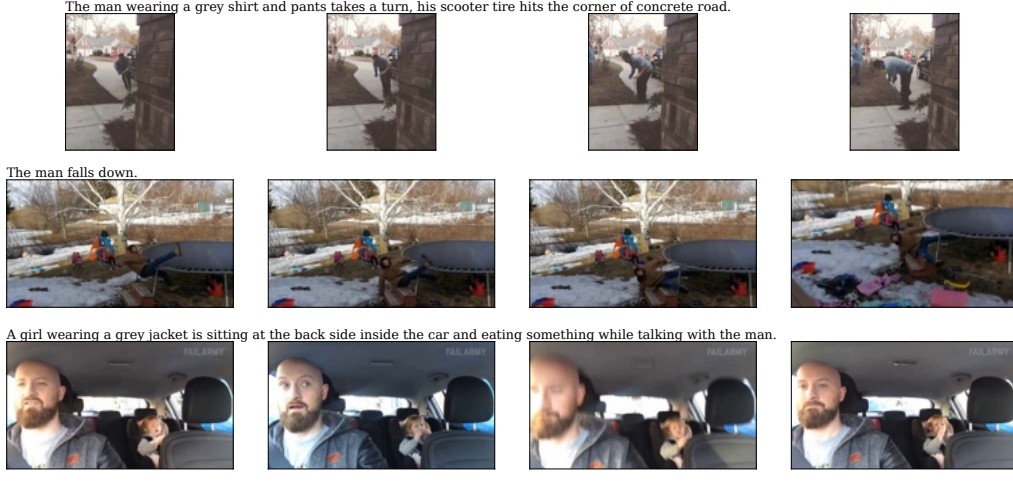

Figure 31: Discarded samples due to video duration.

The boy takes the tissue paper.

A man wearing a yellow outfit is sitting.

The man is watching in the right direction.

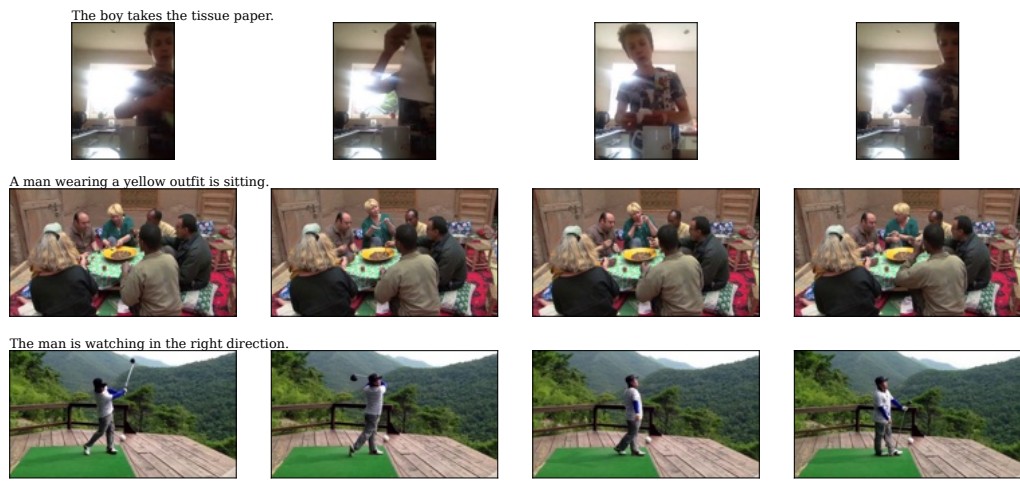

Figure 32: Discarded samples due to low video–text alignment.

The camera still focuses on the man and woman that are singing and playing violin beside her.

singer on the right walks back from the mic, re-approaches it, and then grabs it with their hand.

The camera is recording the group of people on the first boat.

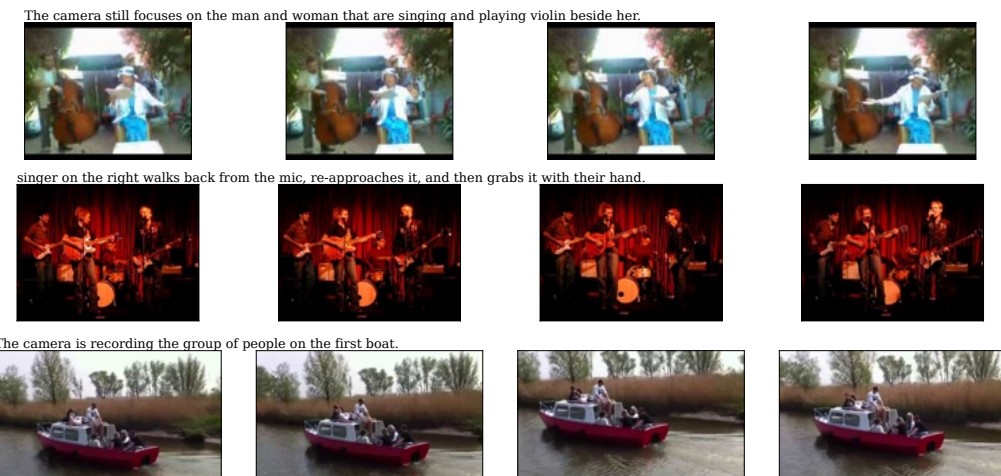

Figure 33: Discarded samples due to low video resolution.

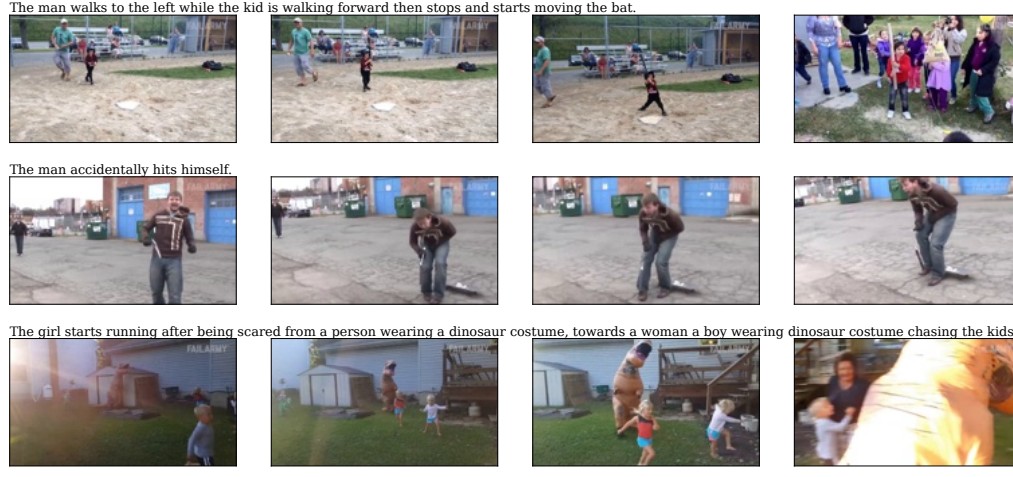

Figure 34: Discarded samples due to manual verification.

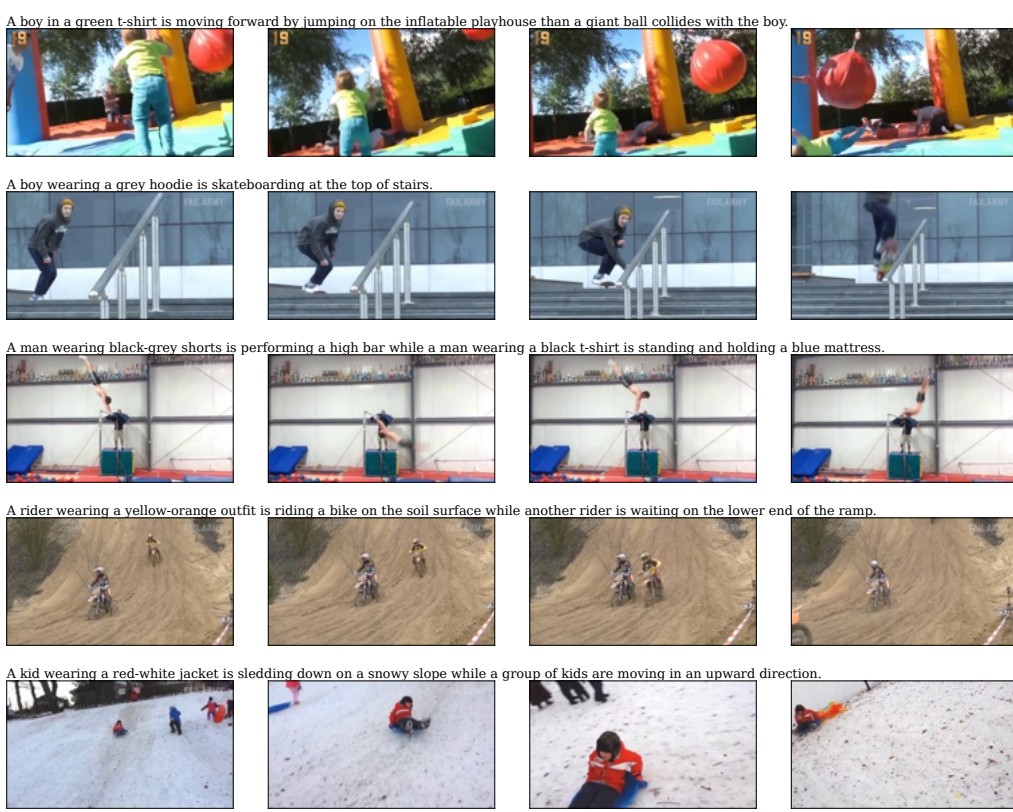

Figure 35: Samples from the final WYD dataset.

BY-NC-SA 4.0 license. Moreover, we use captions and metadata collected in StoryBench Bugliarello et al. (2023), which are also released under a CC BY 4.0 license.

### B.1 DATA FILTERING

High-quality text descriptions are necessary to accurately evaluate text-guided video models. For the datasets above, StoryBench (Bugliarello et al., 2023) includes human-written captions for video generation, as well as useful metadata, including event boundaries and actor identification (*i.e.*, the entities with a key role) in a video (Voigtlaender et al., 2023). WYD leverages StoryBench annotations, but we treat each video segment separately (rather than as part of a sequence) in our

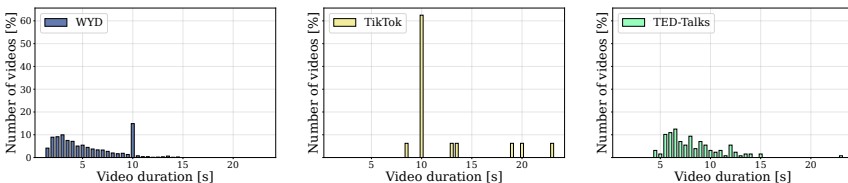

Figure 36: **Distribution of video duration in WYD, TikTok and TED-Talks.** WYD covers actions lasting a few seconds and up to 15s.

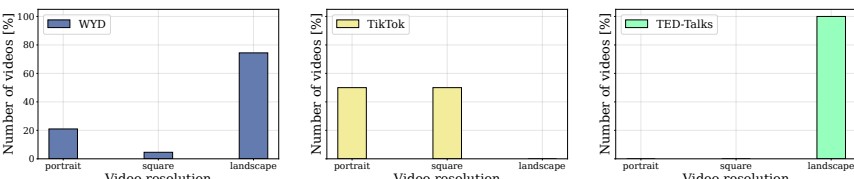

Figure 37: **Distribution of video resolution in WYD, TikTok and TED-Talks.** WYD contains videos that have more diverse aspect ratios.

7-step pipeline (see Fig. 27). Tab. 4 details how each step of our pipeline affected the number of entries of StoryBench, and 28 to 34 show examples that were dropped at each step.

**1. Videos with human actors.** We start by filtering out videos where the main actors are not humans. For Kinetics and Oops videos, we use human-labeled metadata in StoryBench, which associates each caption to its main actor (*e.g.*, "man with white t-shirt"). For DiDeMo, we extract the main actors of each caption with an instruction-tuned LLM (Gemini Team Google, 2023) (note that it is not trivial to use object detectors to extract the main actors due to non-salient humans in a video) with the prompt: "Which living being is performing the main action in the following caption? Reply with one word." From 600+ actors, we manually identify 224 referring to humans, listed in Tab. 5.

**2. Removing scene cuts.** Most of the original videos are single shot, but we found a few of them with multiple scenes. We use a shot detector (Castellano, 2024) and whenever it detects 2–4 cuts, we remove the video if none of the parts lasts for at least 80% of the original duration. Otherwise we replace the original video with that part. Doing so allows us to remove scene cuts while preserving the actions described in the captions.

**3. Ensuring visible humans.** People performing the main action in the video need to be visible, especially in the first frame, to evaluate how image-to-video systems can generate them. We annotate our videos with a state-of-the-art human pose estimator (Yang et al., 2023), and keep only those in which people are 'mostly visible' (defined as 11/18 body keypoints detected) in the first, and at least 70% of the frames.

**4. Removing short and long videos.** Given that our primary goal is to evaluate human video generation, we take into account the capabilities and the computational resources of most existing models to date, and opt to limit videos in our benchmark to a duration between 1.5s and 15s.

**5. High text alignment.** The captions for Kinetics and Oops videos were originally made from the perspective of a specific actor (Voigtlaender et al., 2023). As a result, some captions fail to naturally describe the most salient actors in a video. To minimize such cases, we only keep the actor whose video–caption pairs have the highest similarity according to a fine-grained contrastive VLM (Gao et al., 2022). With the same approach, we then remove the bottom 25% of the videos to maximize text controllability.

**6. Minimum resolution.** We further filter down our data to only include videos with smaller edge of at least 360 pixels, to ensure references of higher quality while keeping enough samples for statistical significance.

**7. Manual verification.** Finally, we meticulously scrutinize the quality of the resulting videos and remove those with (i) significant blur, (ii) poor lighting, (iii) unstable camera, (iv) low motion, (v) unclear captions, or (vi) where the first frame does not capture the main actors. This process, in conjunction with the video categorization below, was done in multiple rounds and took over 250

hours of annotation time. The authors validated the annotations and sought to ensure that the videos had high diversity and would be challenging for current video generation models.

At the end of this pipeline, WYD contains 1,544 high-quality videos (from the original 18,351) which enable the fine-grained analyses described next at a tractable runtime. Figures 36 and 37 show the distribution of video duration and resolution for WYD, TikTok and TED-Talks.

Manual video quality verification and labeling were carried through an extensive period (over two months). Most of the resulting videos (99.5%) have a size of at least 512p. Moreover, we note that a fraction of the frames display motion blur, which is unavoidable in videos with high motion.

## B.2 VIDEO CATEGORIZATION

A key goal of our benchmark is to enable fine-grained understanding of the capabilities of video generation models to synthesize humans across different facets; rather than reporting a single aggregated score, as done with other datasets. To achieve this, we annotate our data with nine categories that capture important aspects for synthesizing videos of humans. Each category in turn contains sub-categories (see Fig. 2 for an overview), each with at least ≈100 samples so as to provide sufficient statistical power for our analyses.

**Number of human actors.** For each video in WYD, we manually label the exact number of humans performing the main actions (*i.e.*, salient for generation), and then group them in three groups (1, 2, 3–8). Notably, each sub-category presents specific challenges, from consistently generating multiple people to more dynamic videos with a single person.

**Human actor size.** The size of human actors can affect how well a video generation model performs. We manually estimate the area covered by the human actors in each video, and categorize them into seven splits of actor size (Fig. 2).

**Human occlusion.** Object consistency is crucial in generated videos, and humans need to keep their appearance despite partial or full occlusions. We measure the average number of body keypoints detected by the pose estimator (Yang et al., 2023), and categorize our videos into five ranges of human actor occlusion (*i.e.*, percentage of keypoints that are not visible).

**Human actions.** The ability to perform a wide range of actions is a distinctive characteristic of humans, and different actions may require disparate generation capabilities (*e.g.*, swimming vs. eating). We manually assign action labels to each video in multiple rounds, adjusting the levels of specificity after each round. This process yields sixteen sub-categories of visually similar actions (as shown in Fig. 2).

**Human locomotion.** We manually classify human body movements into three categories: full-body, partial-body, and hand-focused. While full-body motion indicates actors changing location in the video, partial-body motion involves only moving part of the body (*e.g.*, the arms). We label videos where hands' motion is crucial separately, as existing models often struggle to generate hands (Zhang et al., 2024a; Lu et al., 2024; Lei et al., 2024).

**Camera motion.** We manually label each video as dynamic if the camera follows the actor, or static otherwise.

**Video motion.** The primary aspect that differentiates videos from images is the motion within them. We use an optical flow model (Teed & Deng, 2020) to estimate the amount of motion in each video, which we then use to study how this key aspect affects human video generation across seven motion ranges.

**Actor interactions.** Humans often interact with their environment, either through inanimate objects, animals or other humans. Object interactions are often hard to generate, as they require a deeper understanding of shapes, texture and the physical laws of the world. While previous work only evaluates solo actions (Jafarian & Park, 2021; Siarohin et al., 2021; Chan et al., 2019), our annotations show that most of the videos in our dataset involve interactions.

**Scene.** Different actions are associated with different environments (*e.g.*, swimming) and video generation models should be able to synthesize a variety of environments. We manually annotate the videos in WYD with nine different scenes where actions take place (both indoors and outdoors).

**Discussion.** Fig. 38 shows our UI for video categorization. We find that only a few categories overlap with each other significantly (see Fig. 41 in App. B). Namely, actions and interactions with animals; and video and camera motion, where high-motion videos come from dynamic camera, and videos with low motion correspond to static camera. Interestingly, high-motion videos do not necessarily involve fewer people, but small people are often associated with high motion and full-body movement of a single actor.

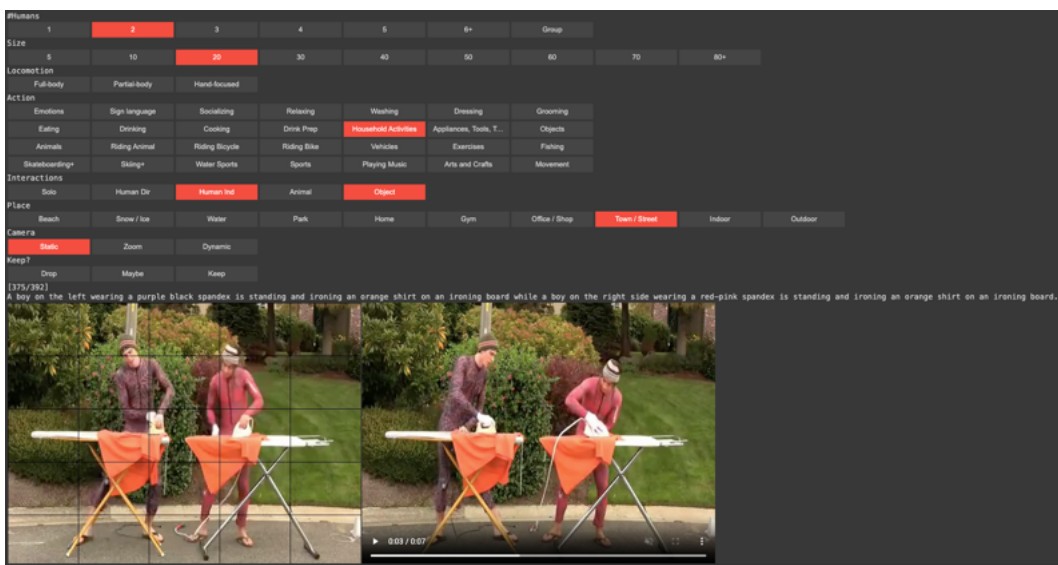

Figure 38: UI used to manually filter and label videos in WYD according to different categories.

### B.3 DENSE ANNOTATIONS

**Video segmentation masks.**    In addition to labeling our videos with 9 categories, we also annotate each human actor in a video with tracked segmentation masks. To do this, we first identify people in the first frame of each video via bounding boxes using OWLv2 (Minderer et al., 2023). After selecting and refining the bounding boxes corresponding to the actors only, we feed them as input to SAM 2 (Ravi et al., 2025), which returns video segmentation tracks for each of the actors. These tracks are further verified and manually corrected at the frame level by the authors, an effort that took over 1000 hours. We use them to define new automatic metrics for WYD by analyzing model performance at the human level, as discussed next. Fig. 39 shows our UI for verifying and fixing video segmentation masks for each actor. Annotations were made at every frame using a brush to extend or delete pixels corresponding to the automatically generated segmentation masks for each human actor.

**Actor 2D pose key-points.**    We further collect human-annotated 2D pose key-points for the (184) actors across 100 randomly selected videos (UI shown in Fig. 40). Human raters are given the option to modify existing keypoints extracted using DWPose (Yang et al., 2023) or to discard them and annotate human skeletons from scratch. This data collection campaign also took over 1000 hours of human rating time. We use these skeletons to verify the trustworthiness of our evaluation framework.

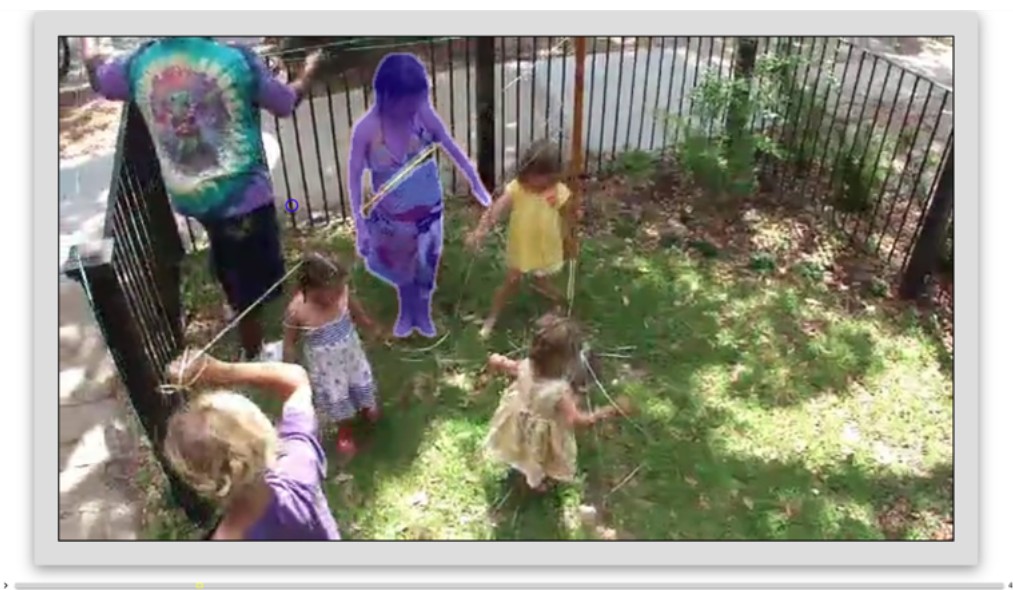

Figure 39: UI used to manually fix video segmentation masks in WYD through a brush to select pixels corresponding to an actor's mask.

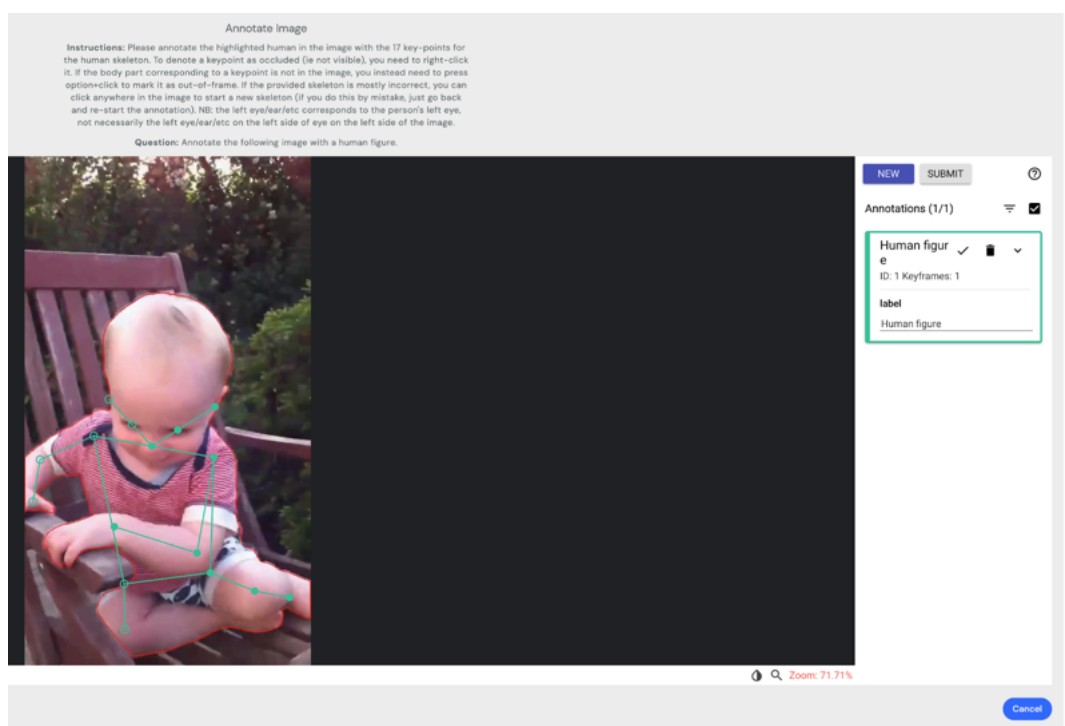

Figure 40: UI used to manually annotate 2D body key-points in WYD.

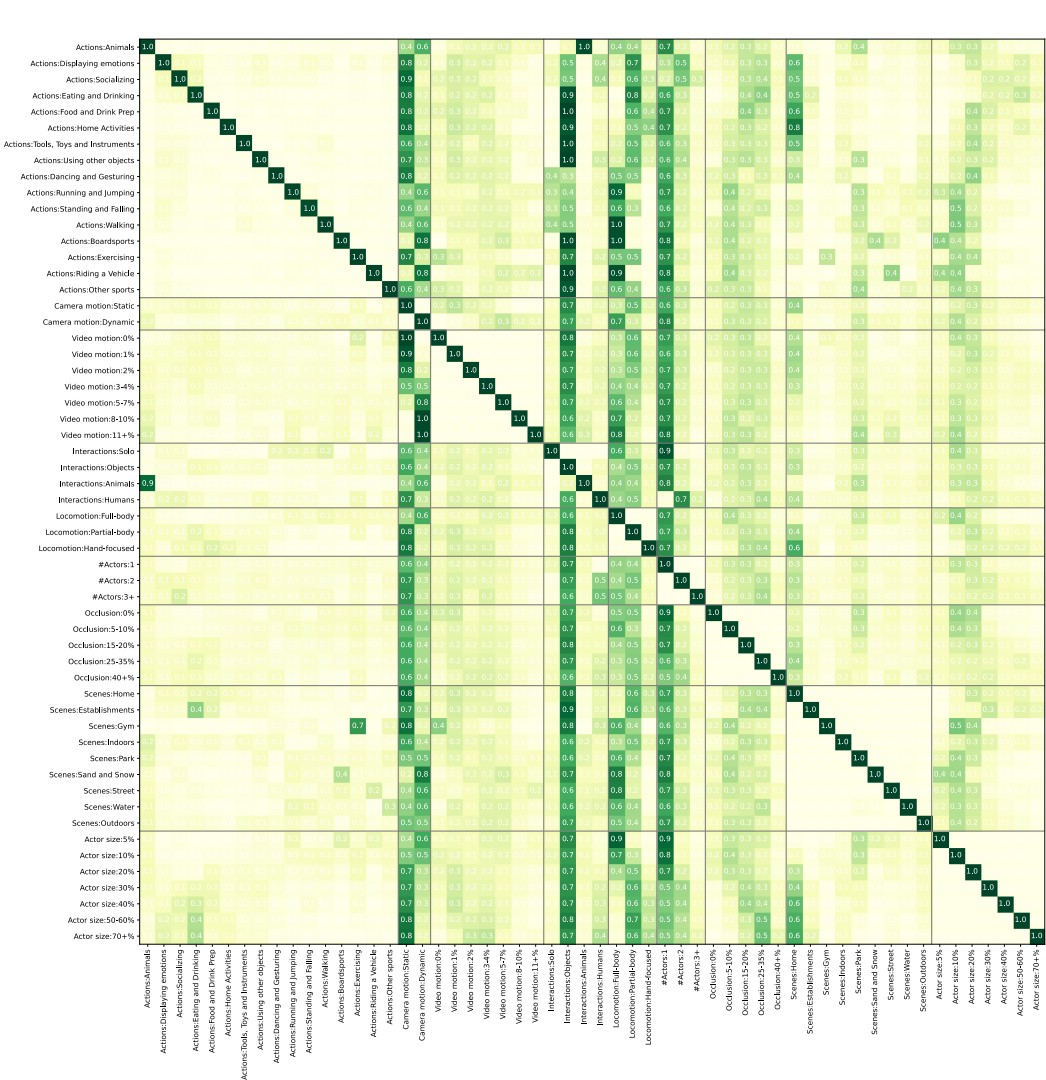

Figure 41: **Percentage of videos overlapping between two categories.** For a given row, computes how many of its videos (in percentage) are also available in another category (column). Best viewed on a screen due to its large number of entries.

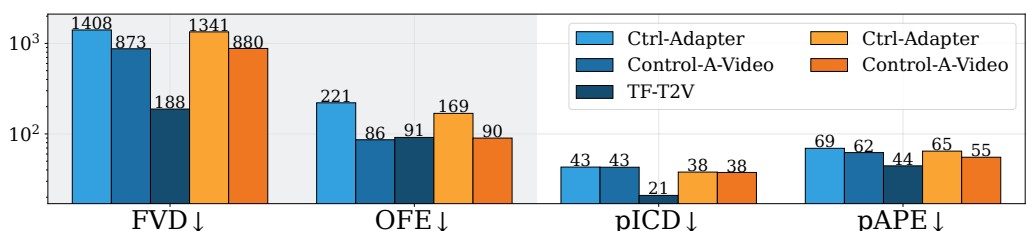

Figure 42: Overall performance (left: video-level, right: human-level) of SOTA controllable depth- and edge-conditioned image-to-video models on WYD_16. Depth models are shown in blue, while edge models in shades of orange. TF-T2V obtains the overall best performance across our models.

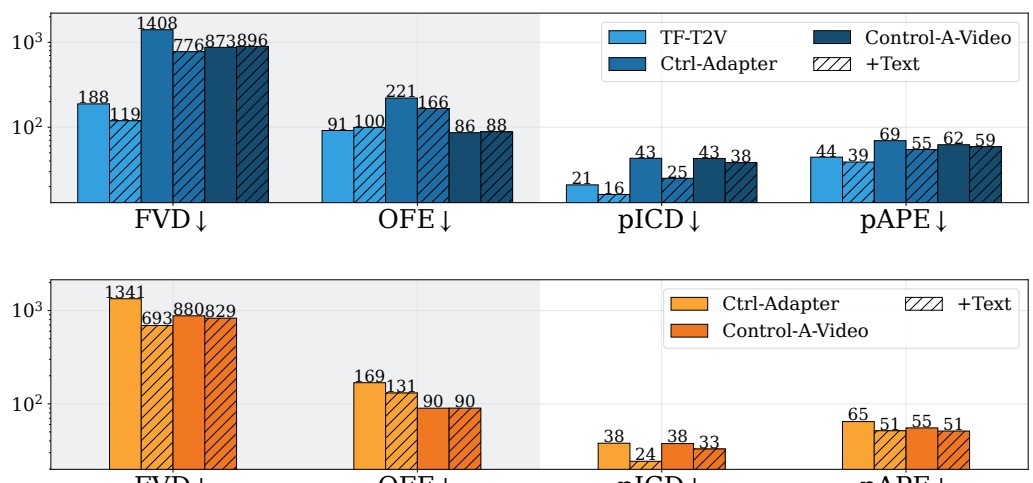

Figure 43: Difference in errors in WYD after adding captions to depth- and edge-conditioned models. Adding text guidance usually improves performance of depth-guided (top) and edge-guided (bottom) models.

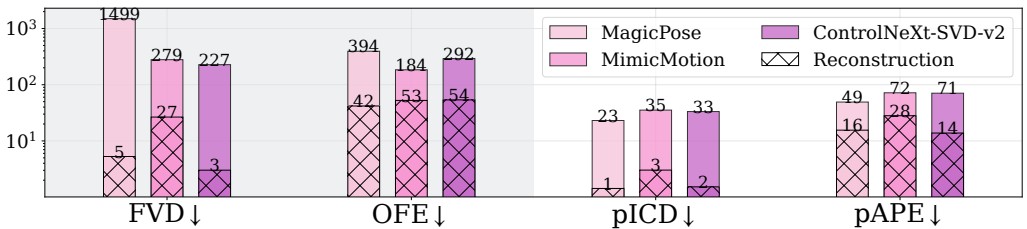

Figure 44: Performance comparison between generation and auto-encoder reconstruction capabilities of pose-conditioned models on WYD_16. We see a clear gap between generation and reconstruction across all metrics, showing that models are capable of generating the reference videos but struggle during the generation process.

## C  ADDITIONAL RESULTS

In this section, we report complementary results from our experiments.

Fig. 42 reports performance for our depth- and edge-conditioned models, while Fig. 43 shows the difference in errors when adding captions as an additional source of guidance to these models.

Figures 44 to 46 report our investigations of pose-conditioned models w.r.t. (i) auto-encoder reconstruction capabilities, (ii) role of different pose detectors, and (iii) the need for human-labeled pose key-points.

Figures 47 to 50 report category-level performance of our best models (MimicMotion, ControlNeXt and TF-T2V) according to sample-level metrics (ICD, OFE, pICD, pAPE).

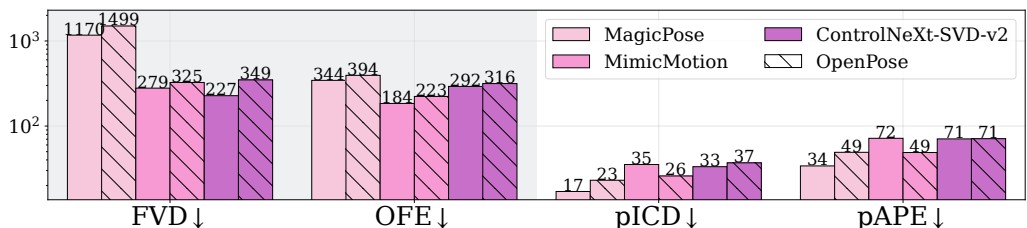

Figure 45: Performance of pose-conditioned models when using the OpenPose detector rather than DWPose (default) on WYD₁₆. We see that using DWPose gives typically better performance, even when applied to MagicPose, which was trained using OpenPose.

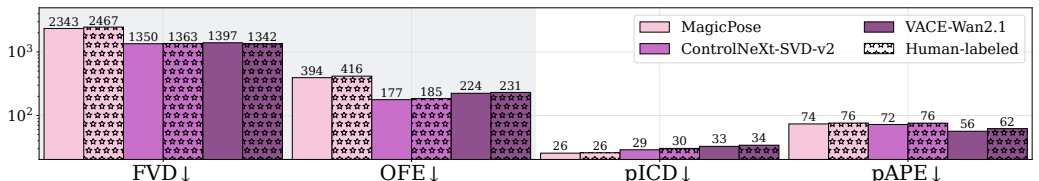

Figure 46: Performance of pose-conditioned models when using the human-labeled 2D body key-points rather than DWPose (default) on a subset of 100 videos from WYD₁₆. We see that models achieve similar performance in these two scenarios, verifying the correctness of our results based on poses detected with DWPose.

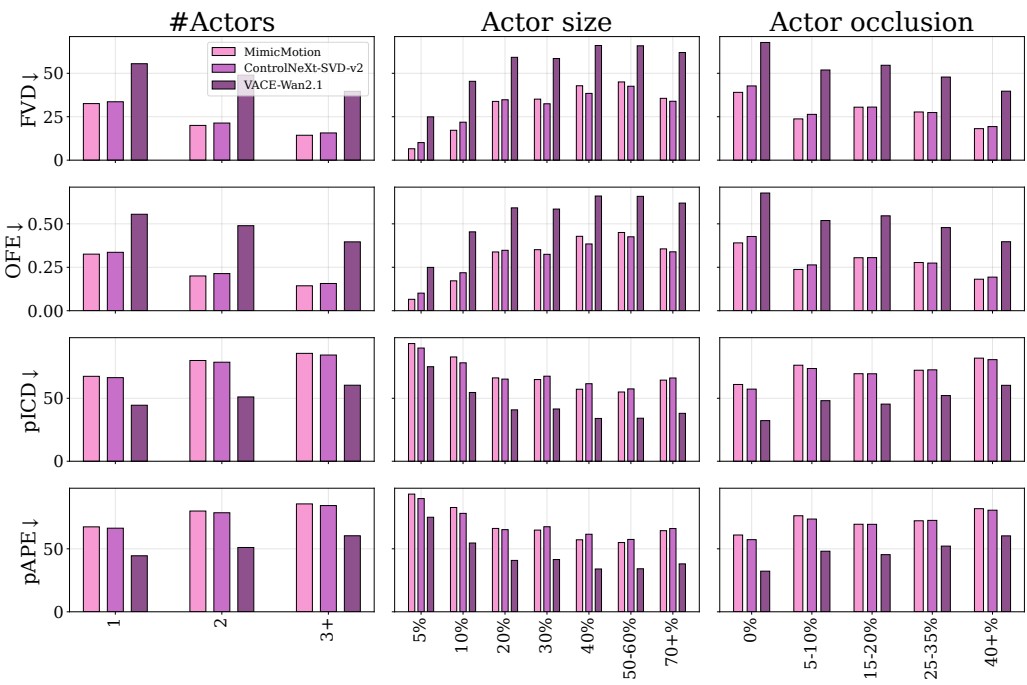

Figure 47: **Performance of best models w.r.t. '#Actors,' 'Actor size' and 'Actor occlusion.'** Animating multiple actors is harder than a single one. Small humans are also harder to generate precisely compared to when they cover a large portion of the frame. Performance also tends to degrade as the amount of occlusion increases.

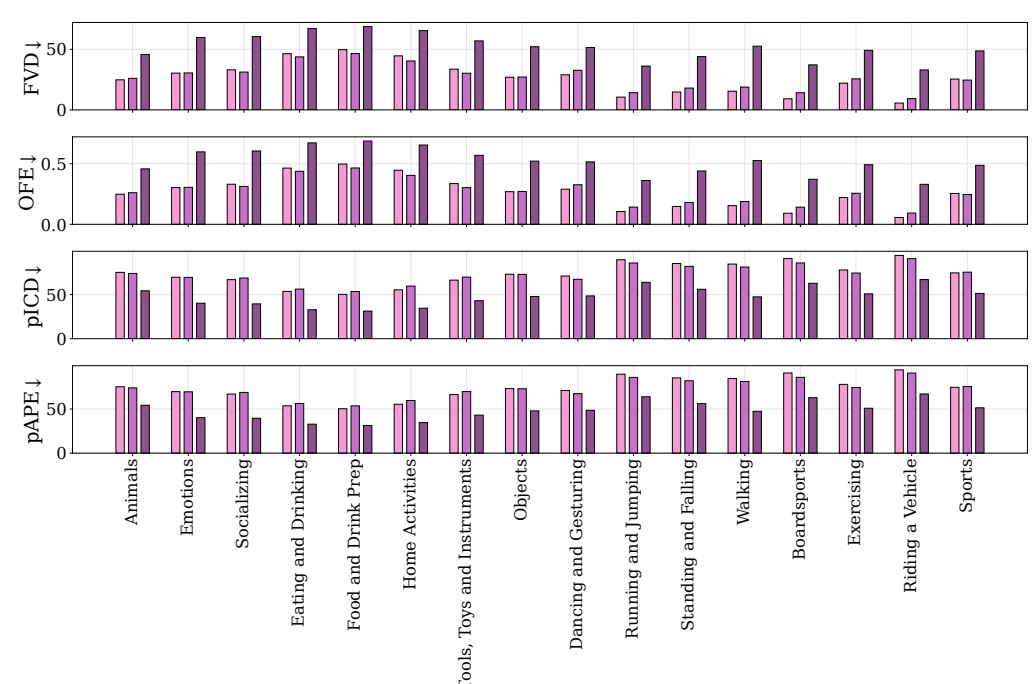

Figure 48: **Performance of best models w.r.t. 'Actions.'** Actions involving animals, riding a vehicle, running and jumping, and boardsports are challenging for SOTA models. Atypical movements, *e.g.*, standing up and falling down, are also hard.

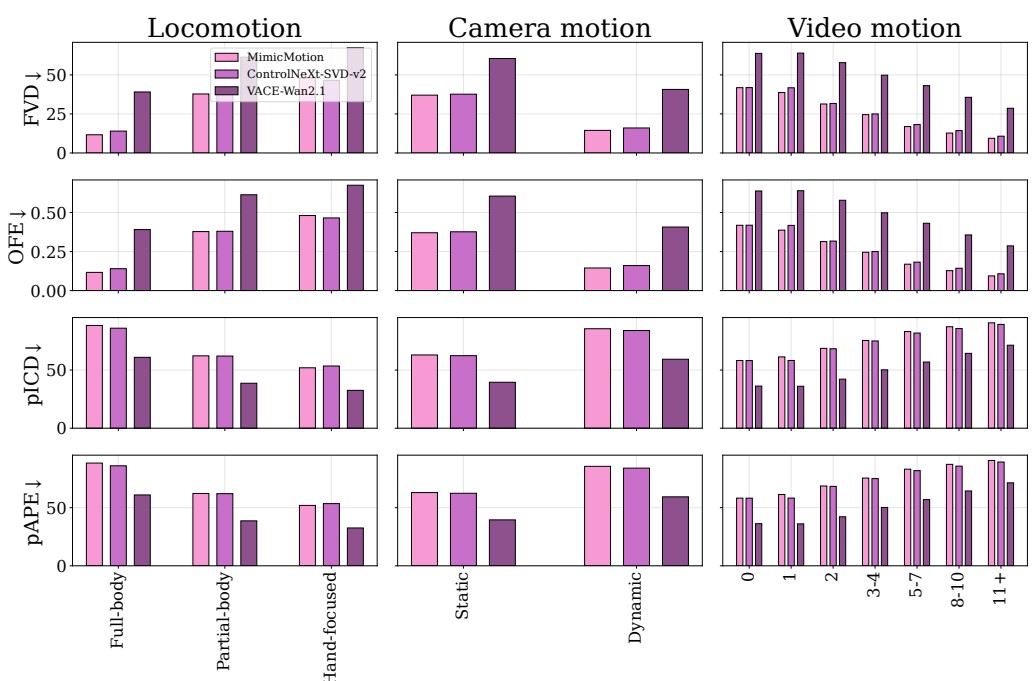

Figure 49: **Performance of best models w.r.t. 'Locomotion,' 'Camera motion' and 'Video motion.'** Videos with full-body locomotion are more challenging to generate due to the larger changes required. Dynamic videos with high levels of motion are more challenging.

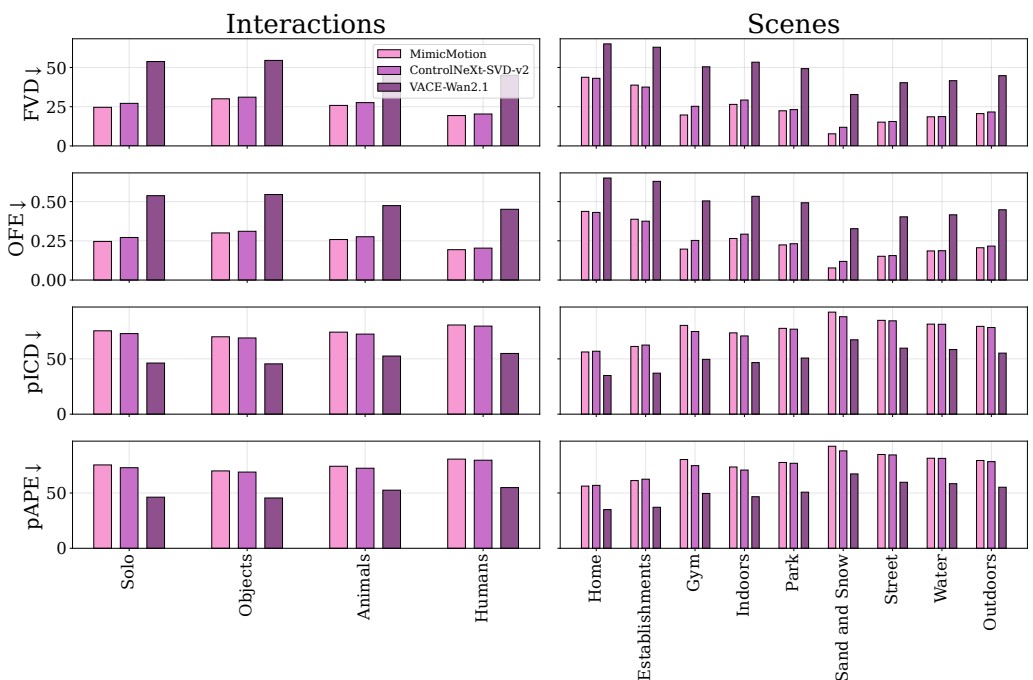

Figure 50: **Performance of best models w.r.t. 'Interactions' and 'Scenes.'** Generating videos of humans interacting with animals or other humans is more difficult than solo videos. Outdoors scenes (*e.g.*, on sand and snow, street, by the water) are also harder for SOTA models.

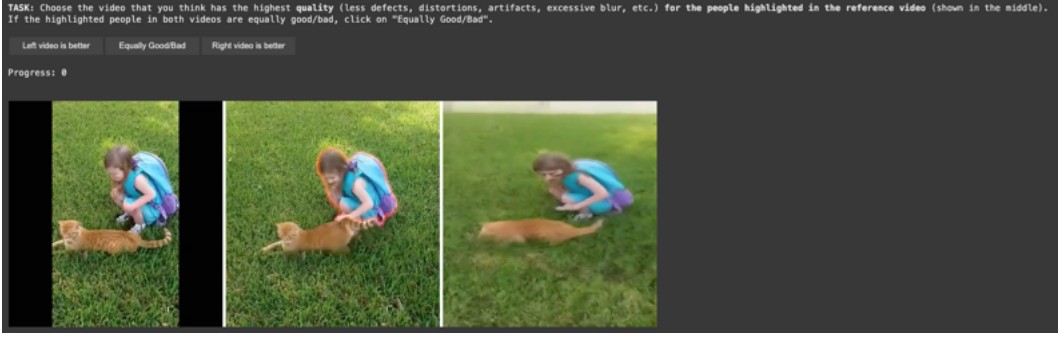

Figure 51: **UI used to collect human preferences in side-by-side evaluations for WYD.** We remove the actors' segmentation masks when comparing video motion similarity. No reference video is shown for video quality comparisons.

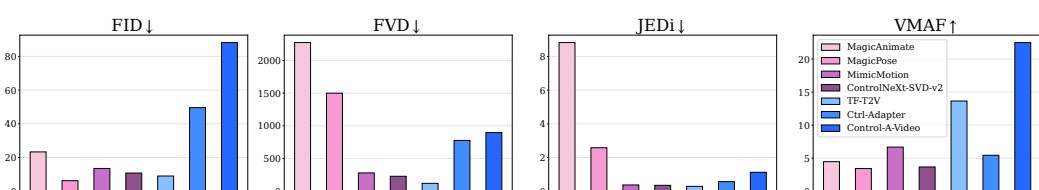

Figure 52: **Comparison of video quality metrics on** WYD₁₆**.** FID favors MagicPose's flickering videos, and VMAF ranks Control-A-Video's videos with distortions and artifacts first. FVD and JEDi rank video generations with high agreement to human judgments.

## D EVALUATIONS

In this section, we report the performance of our evaluated models according to several automatic metrics and detail our human evaluation protocol.

**Automatic metrics.** Figures 52 to 54 show how additional metrics score our seven models w.r.t. overall and human video quality, frame-by-frame similarity and video motion.

We note that, when computing automatic metrics, the pre-processing operations of a given video generation model are also applied to the reference videos. This is because current models were not initially designed to support all aspect ratios, and comparing their generations with the original reference videos would result in unfair evaluations. For instance, ControlNeXt generates portrait videos only. If we had not applied the same transformations to the reference videos, we would not have been able to compute pixel-level metrics. Moreover, this ensures that the results obtained with pAPE capture scale differences (despite model-specific resolutions) and correctly identifies the re-scale and re-center issue in MimicMotion and ControlNeXt described in § 6. We encourage future work to develop models that can process different aspect ratios so as to ensure fully comparable results and benchmarking on WYD.

**Side-by-side human evaluations setup.** For side-by-side evaluation, we sample 100 random videos from WYD and ask four researchers familiar with the task. Each researcher annotates 25 comparisons for each model pair and across four evaluation setups: (i) video quality, (ii) video motion similarity, (iii) human quality w.r.t. reference, and (iv) human motion w.r.t. reference.

For video quality, we compare all 21 model pairs; while we only compare five model pairs (Mimic-Motion vs. ControlNeXt, MagicPose vs. ControlNeXt, MagicPose vs. Ctrl-Adapter, MimicMotion vs. TF-T2V, TF-T2V vs. Control-A-Video) on the other tasks due to the increased amount of time required for careful evaluations.

An example of our UI for side-by-side evaluations is shown in Fig. 51, where we remove the actors' segmentation masks for setup (ii), and only show the generated videos for setup (i). We used the following the templates for evaluation.

1. **Video quality:** Choose the video that you think is of highest quality (less defects, distortions, artifacts, excessive blur, etc.). If both have the same quality, choose the one that is more appealing to you (more interesting, better composition, etc.). If both videos are equally appealing, click on "Equally Good/Bad."

2. **Video motion similarity:** Choose the video that you think best matches the motion of the entire reference video (shown in the middle). Please try to ignore potential defects or bad quality of the videos. If both videos equally follow the motion of the reference video, click on "Equally Good/Bad."

3. **Human quality w.r.t. reference:** Choose the video that you think has the highest quality (less defects, distortions, artifacts, excessive blur, etc.) for the people highlighted in the reference video (shown in the middle). If the highlighted people in both videos are equally good/bad, click on "Equally Good/Bad."

4. **Human motion w.r.t. reference:** Choose the video that you think best matches the movements of the people in the reference video (shown in the middle). Please try to ignore potential defects or bad quality of the videos. If both videos equally follow the motion of the reference video, click on "Equally Good/Bad."

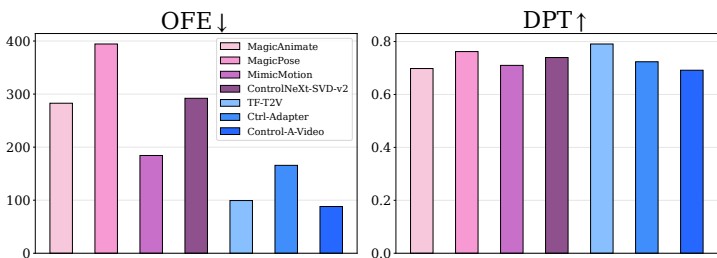

Figure 53: **Comparison of video motion similarity metrics on WYD$_{16}$.** We see that a depth-based metric (DPT) ranks the flickering generations from MagicPose as the second best ones, and those of pose-guided ControlNeXt better than those of depth-conditioned Ctrl-Adapter and Control-A-Video, which better generate videos with dynamic camera. OFE better agrees with the human rankings.

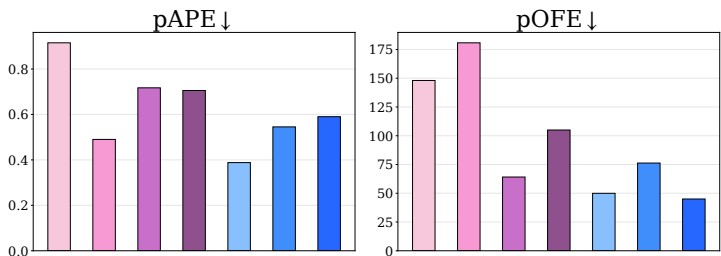

Figure 54: **Comparison of people motion metrics on WYD$_{16}$.** MimicMotion and ControlNeXt do not achieve good pAPE despite good visual quality (FVD). Analyzing further, we find that they always re-scale and re-center the generated humans (see Fig. 11)

## E ETHICS STATEMENT

The aim of *What Are You Doing?* (WYD) is to enable better evaluation of current and future controllable video generation models with respect to human characters and motion, which arguably are of particular importance to people. While this kind of models have great potential to assist and augment human creativity, there are broader societal issues that need to be considered when developing these models.

Video generative models may be misused to generate fake, hateful, explicit or harmful content. For example, they could be used to spread misinformation and portray false situations by synthesizing fake content (*i.e.*, deepfakes). To mitigate these harms, digital watermarks can be applied to generated videos (Luo et al., 2020) to identify whether a given video was produced by a particular model.

Generative models rely on massive amounts of data harvested from the Web, which reflect social stereotypes, oppressive viewpoints, and harmful associations to marginalized identity groups (Birhane & Prabhu, 2021; Birhane et al., 2021; Meister et al., 2022). It is essential that generated content avoids perpetuating harmful stereotypes and respects cultural sensitivities. In fact, models primarily trained on samples with English data may reflect Western cultures (Liu et al., 2021; Pouget et al., 2024). We acknowledge that our benchmark does not explicitly aim to encompass several cultures and populations, and it may perpetuate biases present in the datasets on which it is based. We encourage future work to develop training and evaluation setups that aim to widen the social and cultural representations of these technologies.

Moreover, we note that training video generative models is computationally expensive, both financially (*e.g.*, hardware and electricity) and environmentally, due to the carbon footprint of modern tensor processing hardware. We encourage future research that explores more efficient architectures.

Due to the impacts and limitations described above, we remark that WYD aims to measure progress in video generation research. By no means should our data be extended for use in sensitive domains. We believe that generative technologies, like the type of controllable image-to-video models that can be evaluated in WYD, can become useful tools to enhance human productivity and creativity.

