# OpenReview forum: "What Are You Doing? A Closer Look at Controllable Human Video Generation"
_ICLR.cc/2026/Conference — ICLR 2026 Conference Withdrawn Submission_

### Official Review · Reviewer_RT6L · 2025-10-29

**Soundness:** 3
**Presentation:** 2
**Contribution:** 3
**Rating:** 4
**Confidence:** 3

**Summary:**

The work introduces WYD, a comprehensive and diverse human video generation benchmark that markedly advances prior datasets in scale, variety, granularity, and annotation quality, enabling more rigorous diagnosis across categories such as actors, motion, interaction, and environment. It features a transparent, multi-stage curation pipeline with thorough filtering, annotation, and quality control, delivering high-quality segmentation tracks and pose keypoints. The evaluation protocol combines validated video-level and human-centric metrics and examines their alignment with human judgments. Experiments span multiple state-of-the-art open-source models and conditioning modes (pose, depth, edge, text) with side-by-side comparisons across datasets, revealing weaknesses that earlier work missed. Leveraging WYD’s categorical richness, the analysis exposes interpretable failure modes (e.g., multi-person blending and interaction/scene biases) by dissecting performance across actor count, scale, environment, and motion dynamics. The planned release of the dataset, code, and evaluation framework promotes reproducibility and will help standardize progress in the field.

**Strengths:**

1. Comprehensive and Diverse Dataset: The WYD benchmark significantly improves upon prior human video generation datasets in size, diversity, granularity, and annotation effort, as shown in Figure 2 and throughout Section 3. The range of included categories (actors, motion, interaction, environment, etc.) allows for more rigorous model diagnosis.
2. Fine-Grained Annotation and Curation Pipeline: The paper transparently documents a multi-stage filtering, annotation, and quality-control protocol, expending substantial manual and automatic effort for data validity (see Section 3.1 and 3.3). Segmentation tracks and pose keypoints are of high quality.
3. Holistic Evaluation Protocol: The authors adapt and validate both video-level (e.g., FVD, OFE) and human-centric (pICD, pAPE—see Figure 3) metrics, carefully analyzing their correlation with human judgments (see Section 6 and Table 2).
4. Strong Experimental Design and Systematic Analysis: Evaluation spans multiple state-of-the-art open-source models and various conditioning strategies (pose, depth, edge, text), with side-by-side comparisons on WYD, TikTok, and TED-Talks (Figures 4, 5, and 6). Results are well-visualized, indicating weaknesses not apparent in prior work.

**Weaknesses:**

1. Evaluation on Closed-Source and Broader Models Is Limited: The benchmarking, while well-designed and reproducible, is restricted to open-source controllable models that specifically accept poses, edge, or depth modalities. Given the paper’s broader framing, the exclusion (due to infeasibility but still a limitation) of closed-source or purely text-driven SOTA models may limit WYD’s perceived generality as a universal human-video benchmark. The authors briefly note this, but a more nuanced discussion of these omission’s implications for generalizability and representativeness would strengthen the framing.
2. Metrics—Scope and Limitations: The adapted metrics (pICD, pAPE) focus on human detection/tracking consistency and pose adherence, which are validated against human studies (Section 6, Table 2, and Figure 11). However, significant limitations remain: (a) pAPE depends on single-frame pose detectors, at odds with true sequence coherence; occasional mismatches and pose hallucination are not rare, as admitted. (b) FVD and pICD do not adequately penalize failures at fine semantic details (e.g., missing interaction objects, vanishing animals, or deteriorated faces—see Section 6 and Figure 11). (c) Even with extensive annotation, emotion and facial quality—critical for some applications—are not reliably captured, limiting holistic human video assessment.

**Questions:**

1. Handling and Implications of Segmentation/Pose Estimation Error: The pAPE and pICD metrics depend on existing detectors—how sensitive are these metrics to errors or missed detections in crowded, occluded, or complex scenes (especially for multi-person or dynamic videos)? Have the authors considered or run robustness analyses on label noise, as indicated in some recent pose estimation literature?
2. Dataset Release Ethics and Licensing: What steps are being taken to ensure responsible data release (e.g., subject consent, copyright/cultural sensitivity), considering the source’s heterogeneity and potential identifiability of actors in the videos?
3. Text-to-Video and Closed-Source SOTA Model Evaluation: Are there technical or licensing barriers to evaluating major text-to-video or closed, API-based models? Would the authors consider a leaderboard or challenge format, facilitating external evaluation or submission, as a workaround?

---

### Official Review · Reviewer_iqKE · 2025-10-31

**Soundness:** 2
**Presentation:** 2
**Contribution:** 2
**Rating:** 4
**Confidence:** 5

**Summary:**

This paper introduces "What Are You Doing?" (WYD), a benchmark for fine-grained evaluation of controllable human video generation. Comprising 1,544 videos annotated across 56 sub-categories, WYD addresses the limitations of existing datasets (e.g., TikTok, TED-Talks) by offering greater diversity in actions, interactions, and scenes. The authors propose and validate human-aligned automatic metrics, including pAPE for human movement. Evaluations of seven state-of-the-art models on WYD reveal six systematic limitations previously undetectable, demonstrating its value as a challenging and diagnostic benchmark.

**Strengths:**

The strengths of this paper lie in:
1) The benchmark design and coverage is encouraging. 1,544 curated clips annotated into 9 categories and 56 sub-categories (actors, size, occlusion, actions, locomotion, camera motion, video motion, interactions, scene), enabling stratified diagnosis beyond aggregate scores; >100 samples per sub-category improves statistical power.
2) The description of the data curation pipeline is rigorous, 7-step filtering combining shot detection, pose-visibility thresholding (>=11/18 keypoints in ≥70% frames, visible in first frame), duration bounds (1.5–15 s), min resolution (>=360 px), actor-caption alignment via a fine-grained contrastive VLM, bottom-25% similarity pruning, and extensive manual verification (=250+ hours).
3) The introduced metrics are also contributions. Specifically, introduces pAPE (pose AP error) using DWPose and Hungarian matching for tracking, complementing video-level FVD (I3D), frame-wise ICD (DINOv2), and motion OFE (RAFT). Side-by-side human studies show: FVD/JEDi >> FID for video quality, ICD > SSIM/PSNR/RMSE for frame fidelity, and pAPE > pOFE for human-motion fidelity.
4) The presentation of this paper is good to read, and the methodology straightforward to follow.

**Weaknesses:**

The major concern toward this paper is that: there are already several similar data-sets targeting human-centric video generation, e.g. HumanVid, OpenHumanVid, both of which have already been published on Neurips 2024 and CVPR 2025.
In Table 1, the authors only compare with TikTok and Ted-Talks, both of which are very small-scale and the scenarios are limited, while the follow two with much larger size and more various diversity.

[1] Wang, Zhenzhi, et al. "Humanvid: Demystifying training data for camera-controllable human image animation." Advances in Neural Information Processing Systems 37 (2024): 20111-20131.

[2] Li, Hui, et al. "Openhumanvid: A large-scale high-quality dataset for enhancing human-centric video generation." Proceedings of the Computer Vision and Pattern Recognition Conference. 2025.

Some other technical questions:
1) About constructing validity of pAPE. Using detector-driven keypoints as the primary proxy for human motion fidelity risks misalignment with human perception under camera reframing, scale changes, or detector failures; may incentivize optimization toward detector idiosyncrasies.
2) Are there possible potential training-set leakage? Source datasets (e.g., Kinetics) are widely used to pretrain backbones and generative models; assurances that “wyd videos” were not seen may not preclude semantic overlap, complicating claims about out-of-distribution difficulty.
3) About the human study reporting. Limited detail on raters, sampling, and inter-rater reliability; statistical tests and confidence intervals are not fully specified.
4) About the fairness of system comparison. Auto-regressive extension applied to short-horizon models may disproportionately penalize them; heterogeneous control modalities (pose/depth/edge) assessed under a single protocol; training data differences across models are uncontrolled.
5) How about the metric dependencies and sensitivities? pAPE depends on pose detection stability and penalizes benign re-scaling/re-centering; ICD uses image features without temporal modeling; FVD (I3D) can be resolution/content sensitive and less responsive to small-object errors.
6) How about the dataset and caption biases? Sourced from Kinetics/DiDeMo/Oops with StoryBench captions; heavy filtering and bottom-quartile pruning by similarity may skew distributions toward high text–video alignment.

**Questions:**

Please see the weakness section.

---

### Official Review · Reviewer_Ak8J · 2025-11-01

**Soundness:** 2
**Presentation:** 3
**Contribution:** 3
**Rating:** 6
**Confidence:** 3

**Summary:**

1. This paper introduces "What AreYou Doing?" (WYD), a new, diverse benchmark for controllable human video generation, addressing the limitations of existing datasets like TikTok and TED-Talks.

2. The work provides a rigorously curated dataset of 1,544 videos with 56 fine-grained categories and dense annotations (masks, poses), created via 2500+ hours of manual effort.

3. It also proposes and validates a new evaluation protocol, including human-centric metrics (pICD, pAPE), and uses it to diagnose systemic failures in 8 SOTA open-source models (e.g., struggles with multi-person, dynamic scenes, and object interaction).

**Strengths:**

1. The paper addresses a critical and timely need for a more challenging benchmark as video generation models rapidly advance beyond simple "dancing" or "talking" videos.

2. The dataset construction is meticulous, and the proposed metrics (FVD, pICD, pAPE) are impressively validated against human preferences, demonstrating their reliability over common metrics like FID.

3. The value lies in providing actionable insights, not just scores. The fine-grained category analysis (Fig. 7) clearly identifies specific failure modes (e.g., multi-person, animal interaction) for future research.

**Weaknesses:**

1. The dataset is sourced from Kinetics, DiDeMo, and Oops, which may introduce some bias in terms of culture, demographics, and activities, limiting its universality.

2. The evaluation is restricted to open-source models, meaning the most powerful closed-source models are not benchmarked, confining the conclusions to the "open-source SOTA."

**Questions:**

1. The paper notes pAPE is sensitive to scaling/centering (Fig. 11), while humans are not. Does this make pAPE a flawed metric for perceptual quality, as it may unfairly penalize models that are perceptually accurate but geometrically different?

2. The pAPE metric relies on a pose estimator (DWPose) running on generated videos. How robust is DWPose to the artifacts, blur, or incoherence common in generated content? Its failure could corrupt the metric itself.

3. Given the dataset's focus on interaction (with "Objects" and "Tools"), why not propose a specific metric for human-object consistency (e.g., using object masks) instead of relying on global metrics like FVD that fail to catch "disappearing objects"?

---

### Official Review · Reviewer_p4bN · 2025-11-01

**Soundness:** 2
**Presentation:** 3
**Contribution:** 3
**Rating:** 4
**Confidence:** 3

**Summary:**

The paper proposes a new benchmark for fine-grained evaluation of controllable image-to-video generation of humans.

**Strengths:**

1.  The author proposes a new benchmark for fine-grained evaluation of controllable image-to-video generation of humans. The task is important.

2. The paper is well written

**Weaknesses:**

1.The key point of this paper lies in accurate evaluation. However, the evaluation metrics used are all existing ones, such as FVD, which I find to be quite random. Additionally, I believe Person consistency cannot provide detailed ID consistency evaluation—it can only assess rough appearances, while the real challenge in video generation lies in maintaining consistency in fine details. I think for the paper to have true evaluation value, the focus should be on improving the accuracy of these evaluation models through training, rather than relying on existing and inaccurate evaluation models.


2. The author uses DWPose to evaluate human movement. From my understanding of this model, if there are multiple moving people in the scene, the model’s extraction results may become confused. Therefore, the author’s method can only evaluate simple movement scenarios, and it lacks the ability to evaluate truly complex movement scenes.


3. 100 evaluation samples per category is still a bit small. I think the author could increase the sample size to reduce the randomness in model evaluation.

**Questions:**

see weakness

---

### Note · Authors · 2025-11-14

I have read and agree with the venue's withdrawal policy on behalf of myself and my co-authors.